# Goal-oriented representations in the human hippocampus during planning and navigation

Jordan Crivelli-Decker [1,2] ✉, Alex Clarke [3], Seongmin A. Park [1,4], Derek J. Huffman [1,5], Erie D. Boorman[1,3] & Charan Ranganath [1,2]

Recent work in cognitive and systems neuroscience has suggested that the hippocampus might support planning, imagination, and navigation by forming cognitive maps that capture the abstract structure of physical spaces, tasks, and situations. Navigation involves disambiguating similar contexts, and the planning and execution of a sequence of decisions to reach a goal. Here, we examine hippocampal activity patterns in humans during a goal-directed navigation task to investigate how contextual and goal information are incorporated in the construction and execution of navigational plans. During planning, hippocampal pattern similarity is enhanced across routes that share a context and a goal. During navigation, we observe prospective activation in the hippocampus that reflects the retrieval of pattern information related to a key-decision point. These results suggest that, rather than simply representing overlapping associations or state transitions, hippocampal activity patterns are shaped by context and goals.

Every day, people need to plan and execute actions in order to get what they want. Spatial navigation, for instance, requires one to pull up a mental representation of the relationships between different places— i.e., a cognitive map[1]—and generate a plan for how to reach a goal. Tolman[1] proposed that cognitive maps enable behavioral flexibility so that the same underlying representation can be used to reach different goals. For example, if we wanted to navigate to the Tiger exhibit at the San Diego Zoo we might use the same map-like representation to find the Zebra exhibit.

Several lines of evidence suggest that the hippocampus plays a key role in navigation, though its role in navigation is fundamentally unclear. For example, based on findings showing that hippocampal place cells encode specific locations within a spatial context, many have argued that the hippocampus forms a cognitive map of physical space[2,3]. It is now clear that the hippocampus also tracks distances in abstract state spaces[4–6], potentially supporting the broader idea that the hippocampus encodes a memory space[7] that maps the systematic

relationships between any behaviorally relevant variables[8–10] (see[11,12] for alternative views).

Building on this idea, some have proposed that the hippocampus encodes a predictive map that specifies not only one's current location, but also states or locations that could be encountered in the future[9,13]. For example, the successor representation[9,14], a popular computational implementation of the predictive map model, has been used to argue that the hippocampus represents each state in terms of its possible transitions to future states. This model demonstrates that via an incremental learning process about state-to-state transitions, analogous to model-free learning about rewards, enables organisms to rapidly learn how a sequence of actions can lead to a desired outcome.

Although numerous studies have investigated representations of abstract state spaces in the human hippocampus, two fundamental questions remain unanswered. One key issue concerns the role of context. Single-unit recording studies have reported that the spatial selectivity of place cells is context-specific—that is, the spatial selec-

[1]Center for Neuroscience, University of California, Davis, CA, USA. [2]Department of Psychology, University of California, Davis, CA, USA. [3]Department of Psychology, University of Cambridge, Cambridge, UK. [4]Center for Mind and Brain, University of California, Davis, CA, USA. [5]Department of Psychology, Colby College, Waterville, ME, USA. ✉e-mail: jecrivellidecker@ucdavis.edu

tivity of a given cell in one environment varies when an animal is moved to a different, but topographically similar environment[2,15–19]; see[20] for review). Just as one might pull up different cognitive maps for different physical contexts, it is reasonable to think that we might utilize context-specific maps of abstract state spaces. Computational models have been proposed to explain how the hippocampus might recognize contexts[21–23], but there is little empirical evidence showing whether or how the context in abstract spaces is encoded by the hippocampus.

A second key issue that has not been addressed is the importance of goals in hippocampal representations of abstract task states. Theories of state space representation by the hippocampus rely heavily on results from studies that examined activity in hippocampal place cells during random movements through an environment[18]. Accordingly, studies of abstract spaces in humans typically investigate incidental learning of stimulus dimensions or arbitrary state dynamics[24–26]. These kinds of passive, incidental learning tasks differ from those used by Tolman[1] to demonstrate that animals actively use a spatial representation to guide navigation to particular goal locations in an environment. If the human hippocampus forms an abstract cognitive or predictive map, one would expect to see such a representation during planning and navigation towards different goals in the same context.

Based on what is known from studies of spatial navigation, there is reason to think that hippocampal representations in the context of goal-directed navigation might fundamentally differ from what is seen during random or incidental behavior. For example, hippocampal place cells have differential firing fields during planning depending on the future goal of the animal[27–30], and goal locations tend to be overrepresented[31,32]. Consistent with these findings, fMRI studies of spatial navigation have found that hippocampal activity is modulated by a participant's distance from a goal location[33,34], and that hippocampal activity patterns during route planning carry information about prospective goal locations in a virtual space[35]. These findings suggest that hippocampal representations during planning or navigation in abstract state spaces might be influenced by goals. If this is indeed the case, it would potentially challenge models proposing that the hippocampus encodes a relatively static map of current[2] or possible future states[9].

In the present study, we use functional magnetic resonance imaging (fMRI) to investigate how contexts and goals shape hippocampal representations during planning and navigation (Fig. 1). We devise a task in which participants are required to generate a plan and navigate through two abstract state-space contexts in order to reach a goal state. Critically, the contexts include the same stimuli, with different action relationships in each context. This allows us to examine the impact of context and goals during planning and navigation across perceptually similar sequences. We compare activity patterns elicited during planning of sequences that share a goal to those that had different goals, in order to disentangle the unique contribution of goal information on hippocampal activity patterns. Finally, we analyze the time course of hippocampal patterns while participants actively navigate during the task to examine if current and future states are reactivated in a way that is consistent with computational models of hippocampal function. We show that, during planning, hippocampal representations carry context-specific information about individual sequences towards a goal. Similarly, during navigation, we find prospective activation in the hippocampus that reflects the retrieval of pattern information related to a key decision point. Taken together, our results suggest that hippocampal activity patterns reflect integrated representations of sequences that lead to the same goal. Furthermore, our data support the notion that the hippocampus plays a phasic role in the activation of patterns that contain information about future states by prioritizing sub-goal information during active navigation.

## Results
### Navigating an abstract spatiotemporal map
Prior to scanning, participants were trained to criterion (85% accuracy) to navigate to four goal animals in two distinct contexts that consisted of animals that were systematically linked in a deterministic sequence structure (see Methods). Each zoo context consisted of the same nine animals arranged in a plus maze topology, but the relationships between animals across the two zoos were mirror-reversed and then rotated counterclockwise by 90 degrees (Fig. 1a). At each animal, participants were able to make one of four button presses that allowed them to transition between animals. In the scanner, participants were asked to use their knowledge of the zoo contexts to actively navigate from a start animal to a goal animal (Fig. 1b), where start and goal animals were always at the ends of the maze arms. Each trial consisted of a planning phase and a navigation phase. During the planning phase, a cue indicated the start and goal animals. Next, during the navigation phase, participants saw the start animal alone before moving through a sequence of animals to reach the goal animal. For each animal, participants had to decide which direction in the plus maze to move to

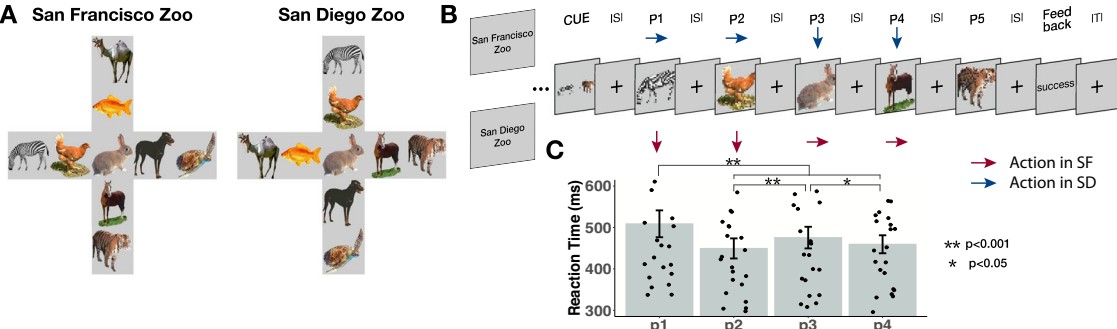

**Fig. 1 | Task design and behavioral results. A** Overhead view of virtual environments. Each context had the same visual information but the specific spatial orientation was mirror reversed and then rotated counter clockwise 90 degrees. This manipulation meant that the action sequence to reach a goal was different across contexts but participants viewed the same visual stimuli. **B** Example navigation trial in the scanner. Participants were first cued with a start and goal location and navigated through the maze one animal at a time. Inter-stimulus interval (ISI) was 3 s. Arrows in red and blue indicate that participants had to make different actions to the same stimuli across contexts to reach their goal during navigation. **C** Group level behavioral results ($N = 23$) from scanner showing elevated reaction times at decision points (Position 1 and Position 3). p1 > p2: $z = 13.97$, $p < 0.0001$, $d = 0.43$, 95% CI [0.29, 0.56]; p1 > p3: $z = 9.13$, $p < 0.0001$, $d = 0.28$, 95% CI [0.18, 0.38]; p1 > p4, $z = 11.67$ $p < 0.0001$, $d = 0.36$, 95% CI [0.24, 0.47]; p3 > p2, $z = 4.84$, $p < 0.0001$, $d = 0.15$, 95% CI [0.074, 0.22]; p3 > p4, $z = 2.536$, $p = 0.0112$, $d = 0.077$, 95% CI [0.014, 0.14]; two-tailed, uncorrected. Pairwise comparisons were conducted using linear contrasts between estimated marginal means (z-test). Error bars represent ± SEM. *$p < 0.05$, **$p < 0.001$. ITI = Interstimulus interval; SF = San Francisco; SD = San Diego. Images used in panels A and B are Copyright 2022, Jordan Crivelli-Decker, and licensors. All rights reserved.

ultimately reach the goal animal. On any given trial, participants were only allowed four moves to navigate to the goal animal and the inter-stimulus interval was fixed to ensure that an equal amount of time was spent at each state. In each zoo context, participants planned and navigated 12 distinct sequences (each repeated 4 times across 6 runs of scanning). In addition, one trial from each sequence was randomly chosen to end early at the rabbit (Catch Trials). This resulted in 72 sequences that could be analyzed (see Methods).

Participants were highly accurate at navigating to the goal animal in each context (Context 1: Mean = 93.7%, SD = 12.9%, Context 2: Mean = 94.7%, SD = 12.2%), with no significant differences in accuracy between contexts ($t_{22}$ = 1.16, $p$ = 0.26, $d$ = 0.24, 95% CI [−0.027, 0.0076]). This suggests that participants had successfully formed distinct representations of each zoo context. We next tested whether participants' reaction times would be modulated by differences in the decision-making demands at different locations in the virtual maze. Specifically, our task was structured such that participants were required to initiate their navigation plan at the onset of the start animal (i.e., position one), and at position three – the center of the plus maze, they needed to choose the correct move in order to reach the goal. Accordingly, we expected reaction times (RTs) to be higher at these positions in the navigational sequence than at other positions. Consistent with this prediction, analyses with a linear mixed effects model revealed a significant effect of position ($\chi^2$(3, $N$ = 23) = 220.99, p < 0.0001, $\eta_p^2$ = 0.03, 95% CI [0.03, 0.04]), such that RTs were elevated at position one and position three, relative to other positions (p1 > p2: $z$ = 13.97, p < 0.0001, $d$ = 0.43, 95% CI [0.29, 0.56]; p1 > p3: $z$ = 9.13, p < 0.0001, $d$ = 0.28, 95% CI [0.18, 0.38]; p1 > p4, $z$ = 11.67 $p$ < 0.0001, $d$ = 0.36, 95% CI [0.24, 0.47]; p3 > p2, $z$ = 4.84, $p$ < 0.0001, $d$ = 0.15, 95% CI [0.074, 0.22]; p3 > p4, $z$ = 2.536, $p$ = 0.0112, $d$ = 0.077, 95% CI [0.014, 0.14]) (Fig. 1). This shows that decision-making demands at key locations, such as choice points, influenced participants' response time.

## Hippocampus is sensitive to context-specific sequences in abstract spaces

During the planning phase (i.e., when participants were viewing the cues), we expected that participants should retrieve information about the sequence of state-action pairs that led from the start animal to the goal animal. Our first analyses targeted the extent to which hippocampal activity patterns carried information about the context and the planned sequence. To address this question, we extracted hippocampal multi-voxel activity patterns on each cue trial and calculated pattern similarity (Pearson's r) between trial pairs that came from repetitions of the same sequence cue in the same context, and compared those to both trial pairs for sequence cues with different start or end points, and trial pairs for sequence cues that came from the same or different context (Fig. 2a). Importantly, visual information was shared across contexts as the cue only indicated the start and goal animal, not the context, and the same cue was associated with different moves between contexts. In addition, only trials which resulted in participants subsequently making the correct moves towards the goal were included in neural analyses.

To test whether hippocampal activity patterns carried information about the context and the planned sequence, we used a linear mixed effects model[36] with fixed effects of context (same/different) and sequence (same/different), and a random intercept for participants (see Methods for model selection details and Eq. 2) to predict pattern similarity in the hippocampus. We reasoned that, during planning, participants retrieved information about the sequence of states and actions needed to reach the goal. Therefore, we predicted that pattern similarity should be higher for sequences that shared the same state-action pairs. Moreover, we predicted that this effect should be context-specific, as the same sequence across contexts have different state-action pairs. Consistent with this prediction, we found a significant sequence by context interaction (Fig. 2b: $\chi^2$(1, $N$ = 23) = 4.26, $p$ = 0.04,

$\eta_p^2$ = 0.06, 95% CI [0.00, 0.20]). Follow up tests showed that patterns evoked by the same sequence cue in the same context were significantly different than all other trial pairs (same seq. + same cx. > diff. seq. + same cx.: $z$ = 2.77, $p$ = 0.006, $d$ = 0.28, 95% CI [0.065, 0.49]; same seq. + same cx. > same seq. + diff. cx.: $z$ = 2.73, $p$ = 0.006, $d$ = 0.27, 95% CI [0.062, 0.48]; same seq. + same cx. > diff. seq. + diff. cx.: $z$ = 2.61, $p$ = 0.009, d = 0.26, 95% CI [0.050, 0.47]; see Fig. 2b). These results show that hippocampal activity patterns carried information about planned state-action sequences within specific contexts.

## Hippocampal activity patterns reflect future goals during planning

The above analysis demonstrates that hippocampal activity patterns carry context-specific information about planned sequences, but there are reasons to think that hippocampal sequence representations might become more similar under certain circumstances. For instance, if the hippocampus uses predictive maps that carry information about possible future states[9], one might expect similar representations of diverging sequences that share the same starting point but lead to different goals by more heavily weighting the immediate state-action pairs that follow planning (see Methods for successor representation simulation details and Supplemental Fig. 1). On the other hand, it is possible that goals are more heavily weighted during planning[37], in which case we might expect similar representations of converging sequences that lead to the same goal but start at different states. We sought to test these ideas by comparing pattern similarity during cues associated with repetitions of the same sequence, cues associated with converging sequences that shared the same goal state, cues associated with diverging sequences that shared the same start state, and cues associated with sequences that had different start and different goal states (Diff. Start Diff. Goal)(Fig. 2a).

A linear mixed effects model with fixed effects for overlap (same sequence/converging/diverging/diff start + diff goal) and context (same/different) and a random intercept for participant (see Methods for model selection details and Eq. 3) showed a significant context by overlap interaction ($\chi^2$(3, $N$ = 23) = 14.75, $p$ = 0.002, $\eta_p^2$ = 0.09, 95% CI [0.01, 0.17]). (Fig. 2c, d). Follow up tests investigating this significant interaction revealed that, within a context, cues with converging goals had significantly higher pattern similarity than cues with diverging goals ($z$ = 2.19, $p$ = 0.03, $d$ = 0.23, 95% CI [0.014, 0.45]), and same sequence cues had higher pattern similarity than cues with diverging goals ($z$ = 3.49, $p$ = 0.0005, $d$ = 0.37, 95% CI [0.13, 0.60]) and cues with different starts and goals ($z$ = 2.77, $p$ = 0.0056, $d$ = 0.29, 95% CI [0.069, 0.51]). However, converging sequences were not significantly different from the same sequence ($z$ = 1.30, $p$ = 0.194, $d$ = 0.14, 95% CI [−0.35, 0.073]). Between contexts, cues of the same sequence and converging sequences showed significantly higher pattern similarity when in the same context (Same Sequence: $z$ = 2.60, $p$ = 0.0094, $d$ = 0.27, 95% CI [0.052, 0.49]; Converging: $z$ = 2.51, $p$ = 0.012, $d$ = 0.26, 95% CI [0.044, 0.48]). In contrast, diverging sequences showed a different pattern of results such that sequences from different contexts had higher similarity ($z$ = 1.89, $p$ = 0.060, $d$ = 0.20, 95% CI [−0.016, 0.41]). Lastly, sequences with different starting states and goals were not significantly modulated by context ($z$ = 0.430, $p$ = 0.67, $d$ = 0.045, 95% CI [−0.16, 0.25]). In sum, these results show that during planning, representations in the hippocampus are differentiated based on future context-specific goals. This suggests that goals may fundamentally shape representations in hippocampus via shared patterns between sequences that lead to the same goal.

## Differences in pattern information during the cue period cannot be explained by shared motor plans or sensory details

The present results are consistent with the idea that the hippocampus supports the planning of state-action sequences toward a goal. Importantly, our cues were carefully controlled, such that participants

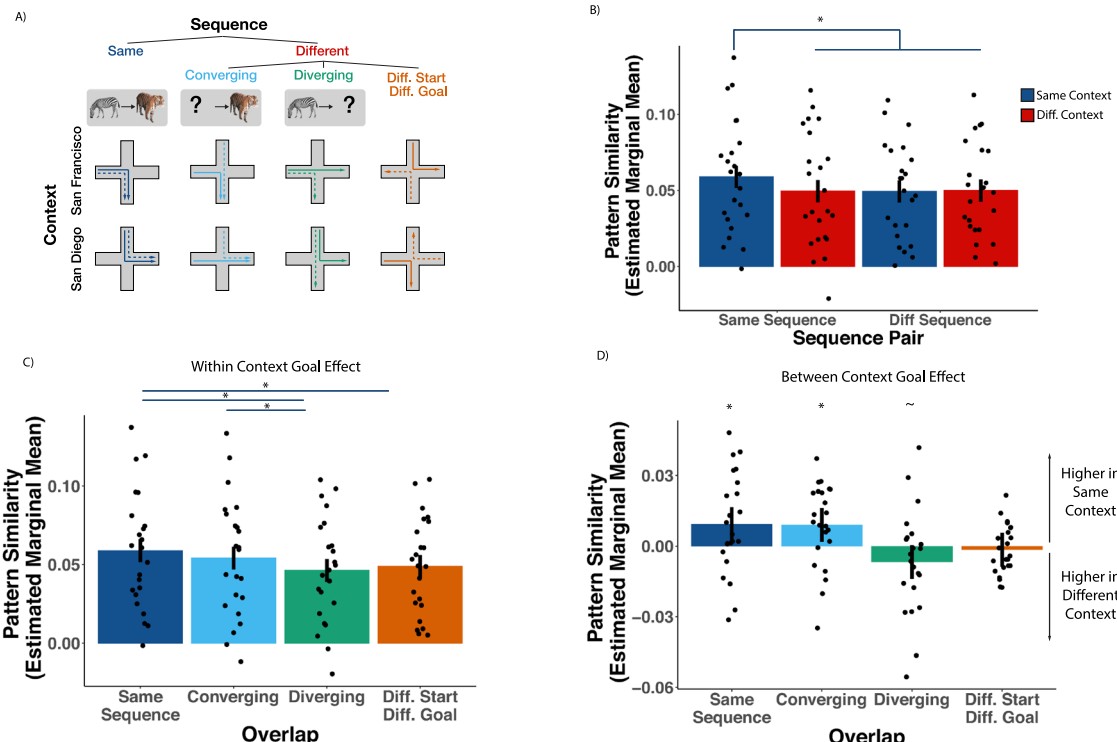

**Fig. 2 | Differential representation of future states in the hippocampus.**
**A** Examples of trial pairs used in pattern similarity analyses during the planning phase. Dashed and solid lines of the same color represent two separate repetitions of the same trial type. **B** Results from bilateral hippocampus. Pairs of trials sharing sequence and context have significantly higher pattern similarity than all other conditions (same seq. + same cx. > diff. seq. + same cx.: $z = 2.77$, $p = 0.006$, $d = 0.28$, 95% CI [0.065, 0.49]; same seq. + same cx. > same seq. + diff. cx.: $z = 2.73$, $p = 0.006$, $d = 0.27$, 95% CI [0.062, 0.48]; same seq. + same cx. > diff. seq. + diff. cx.: $z = 2.61$, $p = 0.009$, $d = 0.26$, 95% CI [0.050, 0.47]; two-tailed, uncorrected). **C** Pattern similarity results comparing converging and diverging sequences within the same context. Same and converging sequences show higher similarity than diverging sequences ($z = 3.49$, $p = 0.0005$, $d = 0.37$, 95% CI [0.13, 0.60]; same seq. > diff. start diff. goal, $z = 2.77$, $p = 0.0056$, $d = 0.29$, 95% CI [0.069, 0.51]; converging > diverging, $p = 0.03$, $d = 0.23$, 95% CI [0.014, 0.45]; two-tailed, uncorrected). **D** Pattern similarity

results displaying the between context goal effect (interaction). Converging and same sequences show higher pattern similarity in the same context. Diverging sequences show higher pattern similarity in different contexts (same seq., $z = 2.60$, $p = 0.0094$, $d = 0.27$, 95% CI [0.052, 0.49]; converging, $z = 2.51$, $p = 0.012$, $d = 0.26$, 95% CI [0.044, 0.48]; diverging, $z = 1.89$, $p = 0.060$, $d = 0.20$, 95% CI [−0.016, 0.41] diff. start diff. goal, $z = 0.430$, $p = 0.67$, $d = 0.045$, 95% CI [−0.16, 0.25]; two-tailed, uncorrected). Pattern similarity was calculated using estimated marginal means obtained from linear mixed effects models. Pairwise comparisons were conducted using linear contrasts between estimated marginal means ($z$-test). Error bars represent 95% confidence intervals of the calculated estimated marginal means. Individual dots represent individual participants mean pattern similarity for each condition. $N = 23$. *$p < 0.05$, ~$p < 0.10$. cx context, seq. sequence. Images used in panel **A** are Copyright 2022, Jordan Crivelli-Decker and licensors. All rights reserved.

viewed visually identical stimuli across contexts and participants did not make responses during the planning phase. However, it is possible that low-level visual representations could be modulated by context[38]. To verify that visual regions did not show any effect of context, we ran a control analysis on an anatomically defined visual cortex ROI (V1/V2). To do this, we compared pattern similarity between cues of the same sequence, cues that had different starting items but the same goal, cues that had the same starting item but diverged to a different goal, and cues that shared neither the start nor the goal. This analysis is identical to the overlap analysis run on hippocampus above (see Methods and Eq. 3 for model details). We found that this visual cortex ROI was only sensitive to visual information (Fig. S2 - Main effect of overlap – $\chi^2(3, N = 23) = 90.24$, p < 0.001, $\eta_p^2 = 0.43$, 95% CI [0.31, 0.52]), and not context ($\chi^2(1 N = 23) = 0.05$, $p = 0.82$, $\eta_p^2 < 0.01$, 95% CI [0.00, 0.03], Interaction: $\chi^2(3, N = 23) = 0.76$, $p = 0.86$, $\eta_p^2 < 0.01$, 95% CI [0.00, 0.02]). This demonstrates that sensory representations of the cue were not modulated by context and likely do not drive any downstream contextual effects observed in the hippocampus.

Having verified that low-level visual information was not modulated by context, we next turned to representations of motor actions during panning. It is conceivable that, during planning, the pattern of results in hippocampus could be driven by overlap in planned movements between converging vs. diverging sequences. To ensure context

effects observed in hippocampus were not due to shared motor information during planning, we examined trial pairs that had the exact same moves, trial pairs that had two moves in common, and pairs that had no moves in common, to ensure that movement information alone was not modulated by context in the hippocampus. Results showed no effect of planned moves or context on pattern similarity (Fig. S2 - main effect of context: $\chi^2(1, N = 23) = 0.46$, $p = 0.5$, $\eta_p^2 < 0.01$, 95% CI [0.00, 0.06]; main effect of move: $\chi^2(2, N = 23) = 1.56$, $p = 0.46$, $\eta_p^2 = 0.01$, 95% CI [0.00, 0.07]; interaction: $\chi^2(2, N = 23) = 2.68$, $p = 0.26$, $\eta_p^2 = 0.02$, 95% CI [0.00, 0.09]).

As a positive control analysis, we also examined an anatomically defined motor cortex ROI (BA4a/4p) to investigate whether we could detect sensorimotor representations and if they were modulated by context information during planning. Results revealed a significant main effect of planned move ($\chi^2(2, N = 23) = 13.95$, p < 0.001, $\eta_p^2 = 0.11$, 95% CI [0.02, 0.23]), and importantly showed that planned movement was not modulated by context (main effect: $\chi^2(1, N = 23), = 0.06$, $p = 0.81$, $\eta_p^2 < 0.01$, 95% CI [0.00 0.04]; Interaction: $\chi^2(2, N = 23), = 0.68$, $p = 0.71$, $\eta_p^2 < 0.01$, 95% CI [0.00, 0.05]; Supplemental Fig. 2) (See Methods and Eq. 4 for model selection details). These results show that our cue period findings in the hippocampus cannot be solely explained by shared motor information of a plan and highlights the role of the hippocampus in retrieving the specific state-

action sequence required to execute a plan. Altogether, these analyses provide an important control and bolster our interpretation of the findings from our analyses of the hippocampus, by showing that primary sensory areas are activating behaviorally-relevant representations during planning, but that the effects of context and goal are only present in hippocampus.

## Representation of behaviorally relevant sequence positions during navigation

Having established that the hippocampus represents information about context-specific goals during planning, our next analyses turned to how state-action information is dynamically represented during navigation. Available evidence suggests at least three ways that navigationally-relevant information might be represented by the hippocampus. Based on classic studies of place cells, we might expect the hippocampus to represent the current state as participants navigated toward the goal. Alternatively, based on predictive map models[9], we could expect that the hippocampus would represent not only the current state but also future states.

A third possibility is that the hippocampus might preferentially represent goal-relevant information during navigation. In our study, the most behaviorally significant points in a navigated sequence were the starting point (position 1), when a goal-directed plan must be initiated, and the center of the maze (position 3), a critical sub-goal where one's decision will determine the ultimate trial outcome. This was confirmed by our behavioral analyses that revealed that participants were slower to respond at positions 1 and 3 (Fig. 1). We therefore reasoned that participants might be likely to prospectively retrieve hippocampal representations of these states during navigation.

To test this prediction, we examined pattern similarity differences during navigation across converging and diverging sequences in the same zoo context. Converging and diverging sequences were chosen because these sequences have an equal number of overlapping states, but the timing of the overlap is systematically different. Both the current state and standard predictive map models would suggest that pattern similarity during navigation should reflect this pure overlap–early in a sequence there should be higher pattern similarity across pairs of diverging sequence trials, and late in a sequence there should be higher pattern similarity across pairs of converging sequence trials. In contrast, a goal-based account would predict that pattern similarity could reflect prospective coding of goal-relevant information[35,39], which should be higher across converging sequences (which share the same upcoming goal), relative to diverging sequences (which overlap in early states but lead to different goals).

We used a time-point by time-point pattern similarity analysis approach that enabled us to examine information in multivoxel activity patterns about current, past, and future states to test our key hypotheses. This technique is conceptually similar to cross-temporal generalization techniques used in pattern classification analyses[40]. First, we extracted the time-series for each navigation sequence using a variant of single trial modeling that utilizes finite impulse response (FIR) functions[41], allowing us to examine activity patterns for each time point (TR) as participants navigated through the sequence of items. Importantly, incorrect trials were excluded from this analysis. As depicted in Fig. 3, we quantified pattern similarity between pairs of navigation sequences (e.g. zebra to tiger sequence compared to the camel to tiger sequence) at different timepoints (e.g., TR 1 to TR 10), which yielded a timepoint-by-timepoint similarity matrix for each

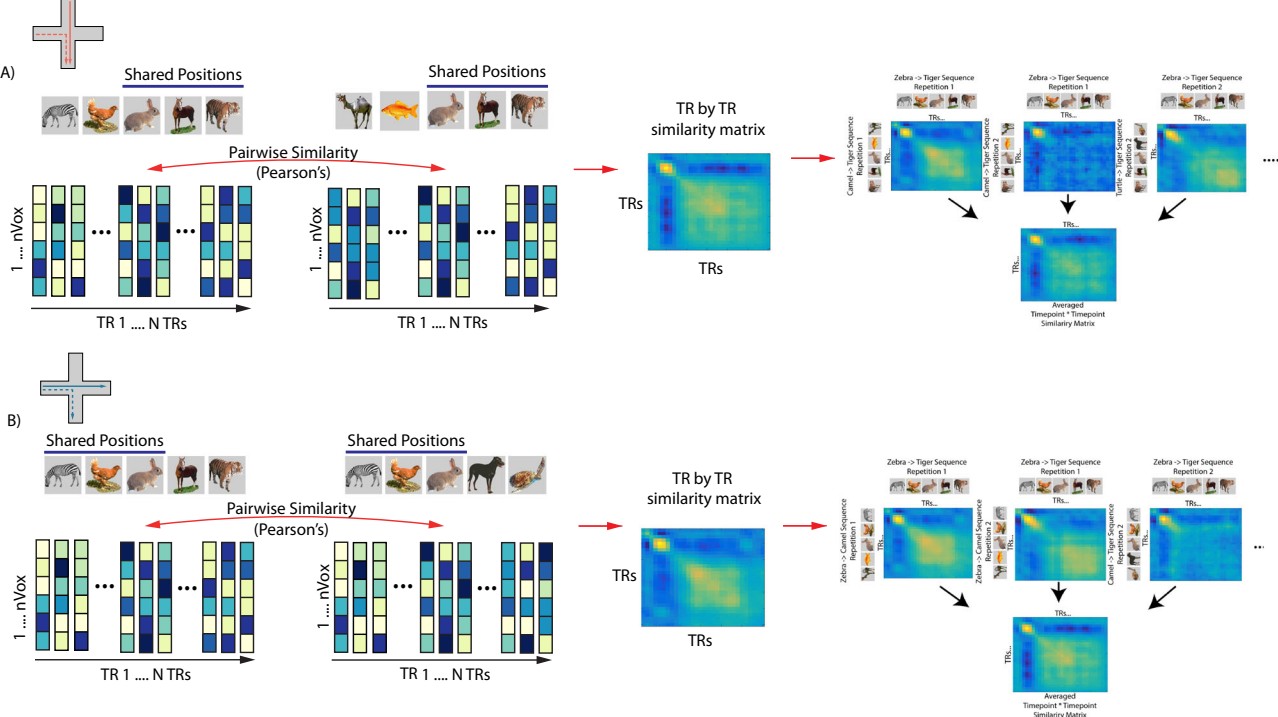

**Fig. 3 | Schematic depiction of procedure to obtain time point by time point similarity matrices. A** (Left) Dashed and solid lines on the maze indicate an example pair of trials correlated. TR by TR spatio-temporal patterns was obtained for a pair of sequences (converging in this example). Pattern similarity was computed between every possible pair of spatial patterns (voxels) over all timepoints (TRs) from a region of interest. (Middle) This procedure yielded a TR by TR similarity matrix for a given sequence pair. Note, that because the sequences are from different repetitions across fMRI scanning runs, the diagonal is not perfectly correlated. (Right) This was repeated for every possible converging sequence pair in the data set. The resultant TR by TR matrices were than averaged to create a participant-level converging TR by TR matrix. Participant-specific averaged TR by TR matrices was then statistically compared to diverging sequences using cluster-based permutation tests (see Methods). **B** Same as **A** but using an example diverging sequence pair. Images used in panels **A** and **B** are Copyright 2022, Jordan Crivelli–Decker and licensors. All rights reserved.

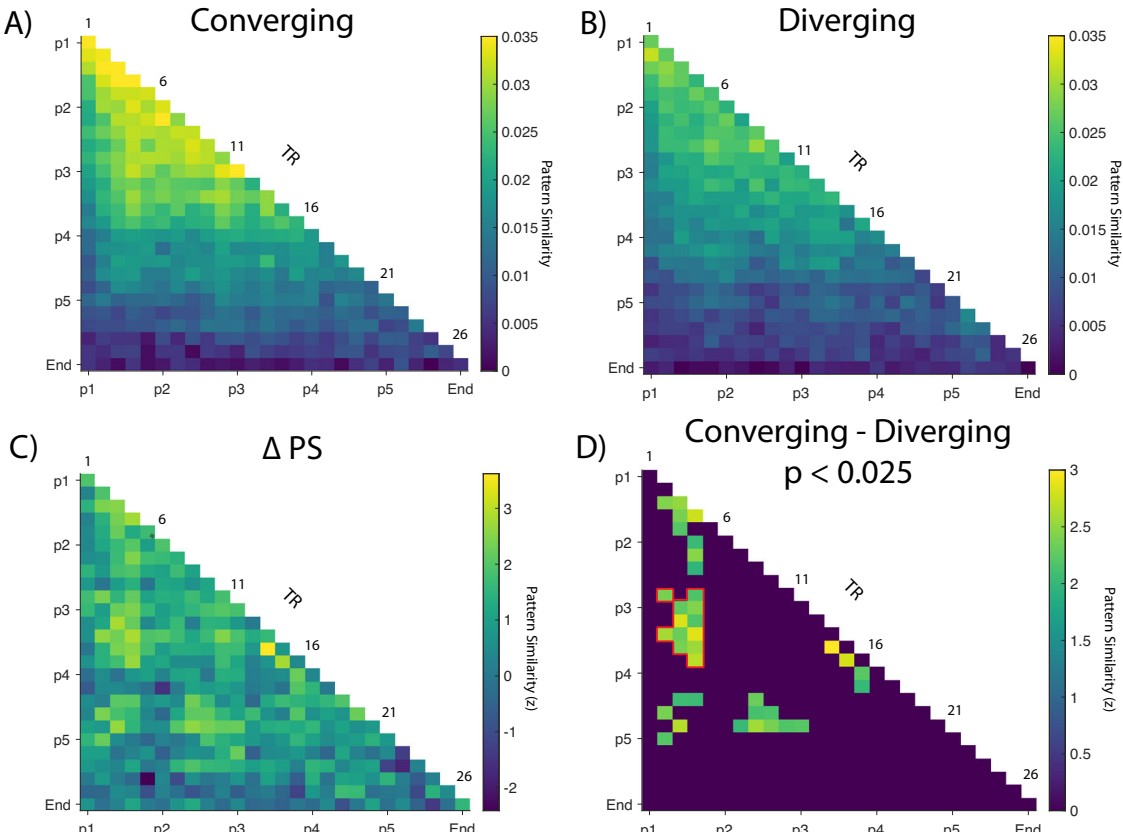

**Fig. 4 | Results from TR by TR pattern similarity analysis during active navigation in bilateral hippocampus. A** Group level pattern similarity results from converging sequences during active navigation. **B** Same as A but showing diverging sequences. **C** TR by TR pattern similarity results depicting a statistical map of converging − diverging. *Z* values were calculated using a bootstrap shuffling procedure with 10,000 permutations. **D** Thresholded statistical map at *p* < 0.025 (two-tailed). Cluster based permutation tests with 10,000 permutations[88] were performed with a cluster defining threshold of *p* < 0.05 (two-tailed) and a cluster alpha of 0.05 (two-tailed). Outlined in red is a significant cluster of timepoints that

survives multiple comparisons correction ($T_{22}$ = 3.34, *p* = 0.038, *d* = 0.27, 95% CI [0.0031, 0.013], maximum cluster corrected). Note that this cluster corresponds to approximately position 1 activating position 3 which was shared by both converging and diverging sequences. Trial labels were manually lagged by 4 TRs (TR = 1.22, Inter-Item-Interval = 5 s) to account for hemodynamic response lag. In panels C and D, each pixel of a statistical comparison (T-value, *N* = 23) was converted into a Z value by normalizing it to the mean and standard error generated from our permutation distributions (see Methods).

condition (converging or diverging sequences). The diagonal elements for this matrix reflect the similarity between pairs of animal items from the same timepoint in the sequence. Off-diagonal elements reflect the similarity between an animal at one timepoint in the sequence and animal items at other timepoints in the sequence. Importantly, incorrect trials were excluded from this analysis.

Separate timepoint-by-timepoint correlation matrices (Pearson's r) were created for pairs of converging sequence trials and pairs of diverging sequence trials. We next computed a difference matrix and tested for statistically significant differences between converging and diverging sequences, correcting for multiple comparisons using cluster-based permutation tests (10,000 permutations, see Methods for more details, for individual subject contrast maps see Supplemental Fig. 3).

As noted above, diverging sequences have overlapping states early in the sequence, and converging sequences have overlapping states late in the sequence. If the hippocampus represents only current states, we would expect to see pattern similarity differences between converging and diverging close to the diagonal of the timepoint-by-timepoint matrices – that is, we would expect higher pattern similarity for diverging pairs during timepoints early in the sequence and higher pattern similarity for converging pairs during timepoints late in the sequence. If the hippocampus represents current and temporally-contiguous states, as suggested by predictive map models, we would expect that at early positions, we would expect higher pattern

similarity for diverging sequences, both on- and off-diagonal, and at late positions, we would expect higher pattern similarity for converging sequences both on- and off-diagonal. Finally, if the hippocampus preferentially represents goal-relevant information during navigation[37,39], we would expect to see higher off-diagonal pattern similarity only for converging sequences, because only converging sequences share the same goal. Specifically, we expected higher off-diagonal pattern similarity between goal states and earlier positions in the sequences.

Consistent with the prospective representation of goal-relevant states in the hippocampus, we found several clusters showing higher similarity for converging compared to diverging sequences (Fig. 4, Supplemental Figs. 5–7). Interestingly, there was a significant off-diagonal cluster (outlined in red: $T_{22}$ = 3.34, *p* = 0.038, d = 0.27, 95% CI [0.0031, 0.013], maximum cluster corrected) that roughly corresponds to the activation of the decision point (position 3) when participants were at position 1 (approx. TRs 10-15). Other clusters tended to overlap with key locations in the experiment, which roughly correspond to position one activating position five (TRs 18 to 21) and position three activating position five (TRs 18 to 20) (Fig. 4D), although these clusters did not survive multiple comparison correction. These data are consistent with the idea that information about position 3 was preferentially activated in converging sequences, in which the same key decision was required to navigate to the same goal.

## Discussion

The aim of the present study was to identify how the hippocampus represents task information during planning and navigation towards a behavioral goal. During planning, we show that hippocampal representations carried context-specific information about individual sequences to a goal. Surprisingly, not all sequences were equally differentiated, such that sequences that converged on a common goal showed higher pattern similarity compared to diverging sequences, despite an equal amount of overlap between the conditions. Similarly, during navigation, we found that the hippocampus prospectively activated goal-specific representations of the key decision point. Taken together, our results suggest that the hippocampus forms integrated representations of sequences that lead to the same goal. Furthermore, they support the notion that the hippocampus plays a phasic role in the activation of patterns that contain information about future states and prioritizes sub-goal information during active navigation. In summary, our data are consistent with the idea that rather than simply representing overlapping associations, hippocampal representations are shaped by context and goals.

### The hippocampus represents context-specific goal information during planning

A key finding from the present study is that, during planning, hippocampal activity patterns are organized such that they either generalize or differentiate between sequences depending on the goal, and do so in a context-specific manner. These findings are relevant to theories which propose that prospective thought (prediction/planning) relies on the same circuitry used for episodic memory[42–44]. In support of this idea, place cells fire in a sequence that represents the path that an animal will take in a phenomenon described as forward replay[45,46]. This work supports the hypothesis that planning may be supported by physiological mechanisms at a single neuron level (but see ref. [47]). Building on this work, Brown et al.[35] used high-resolution fMRI in humans to examine hippocampal activity during goal-directed navigation in a virtual reality (VR) paradigm. Brown et al. demonstrated that, during planning, hippocampal activity patterns could be used to accurately decode future navigation goals, even across different start positions and routes. Thus, their findings demonstrated that fMRI activity patterns in the hippocampus carried information about future navigational goals. Brown et al. interpreted their findings as evidence that the hippocampus supports imagination or mental simulation of a route towards a goal.

Our findings suggest an important constraint on the role of the hippocampus in imagination and simulation. In our study, if participants simulated the sequence of sensory events that led to the goal (i.e., imagining the sequence of animals), we would expect hippocampal representations to generalize across repetitions of the same sequence of animals across the two different zoo contexts. Instead, we found that hippocampal representations during planning were context specific, such that pairs of trials involving the same sequence of animals across different contexts were indistinguishable from entirely different sequences. Moreover, similarity across different sequences that led to the same goal in the same zoo context was indistinguishable from similarity across repetitions of the same sequence in the same context. Thus, in our study, hippocampal activity most likely did not reflect the imagination of a sequence of stimuli per se, or even a specific sequence of states, but rather a context-specific representation of behaviorally relevant points to achieve a goal.

Together with prior research, our results are relevant to an emerging body of work suggesting that goals and other salient locations have a measurable impact on spatial and non-spatial maps in the brain[19,48–51]. For example, McKenzie et al.[19] found that rewarded events had higher pattern similarity within a context compared to unrewarded events. Moreover, there is evidence that, after learning in a reward-based foraging task, place cells tend to be clustered around goal locations[31,32]. This could go some way towards explaining our results of increased pattern similarity for sequences that converge on the same goal. Considering the current work and past findings, we propose that hippocampal representations are flexibly modulated depending on current behavioral demands, incorporating trial-specific information that allows organisms to realize a specific goal[52].

Our findings are also relevant to past work showing that the hippocampus represents information about specific sequences of objects[25,53–57]. Studies examining how the brain represents routes with multiple paths or that are hierarchical in nature show that activity in the hippocampus is higher when planning and navigating an overlapping route and that, during navigation, the univariate bold signal is modulated by distance to a goal[58–60]. In one study, Chanales et al.[60] show that representations of overlapping spatial routes become dissimilar over learning. This is potentially at odds with the current findings, where we find that routes that overlap in their goal show higher pattern similarity compared to routes that do not share a goal. However, participants in Chanales et al. passively viewed pictures along routes, whereas participants in our task actively navigated. As mentioned earlier, rodent studies suggest that hippocampal spatial coding can shift dramatically between goal-directed behavior and random foraging in the same context. Moreover, in Chanales et al. it would make sense for participants to differentiate overlapping routes because they did not include sequences that converged on the same goal. Thus, it would be optimal to learn a unique representation for each spatial route in order to predict the outcome. In contrast, in our experiment, all trials that converged on the same goal required the same key decision at position 3, regardless of the starting point. In this situation, it is optimal to learn a representation that captures the information that is common to any sequence that converges on the same goal. For example, as depicted in Fig. 1, any trial with a tiger as the goal animal will require participants to choose the down button at position 3. In the next section, we explain why results from the navigation period are also consistent with this interpretation.

### Reinstatement of remote timepoints in the hippocampus during navigation

If the hippocampus supports prospective planning for goal-directed navigation, then it is important to understand how it functions when such actions are taken when navigating abstract spaces. For example, if the hippocampus is involved in retrieving the specific state-action plan, what is its function once this plan is executed? To address this question, we contrasted pattern similarity during the navigation phase across pairs of converging sequences against pairs of diverging sequences.

As noted above, the animals in the first three positions overlapped across diverging sequences, whereas the animals in the last three positions overlapped across converging sequences. Thus, if the hippocampus only represented the current state during navigation, we would have expected pattern similarity on the diagonal in Fig. 4 to be higher for diverging trials for early time points, and then higher for converging trials in the later time points (see also Supplemental Fig. 4). If participants solely retrieved past states during navigation, we would expect off-diagonal pattern similarity to be higher for diverging sequences than converging sequences (because the first three positions were common for the diverging sequences). Our data were inconsistent with both of these accounts. Instead, we found that off-diagonal pattern similarity was higher for converging than for diverging trial pairs, suggesting that hippocampal activity patterns carried information about future timepoints during navigation.

The significant cluster of increased pattern similarity for converging, relative to diverging, sequences was consistent with the interpretation that, at the outset of the navigation phase, participants prospectively activated a representation of position 3. This result is notable for two reasons. First, participants were engaged in active, self-

initiated navigation, and as such, we would expect considerable variability in the timing of prospective coding across trials and across participants. The fact that prospective coding of position 3 (as indicated by off-diagonal pattern similarity) was nonetheless reliable across participants attests to the significance of this position to successful task performance. Second, the finding is notable because the stimulus at position 3 is exactly the same for all trials in all contexts. Thus, the disproportionate representation of position 3 across convergent sequences could not solely reflect the identity of the stimulus itself.

As noted above, the correct decision to be made at position 3 depends on one's current goal and context. All converging sequences share the same decision at position 3 because they share the same goal, whereas diverging sequences are associated with different decisions at position 3 because they involve different goal states. These results are consistent with the idea that participants prospectively activated the most goal-relevant information in the upcoming sequence, namely the context- and goal-appropriate decision at position 3.

Consistent with our study, research in rodents shows that hippocampal ensemble activity differs between routes that share a common path but lead to a different goal[28–30,61,62]. There are also findings that demonstrate predictive hippocampal representations that are related to future behavior in both spatial and non-spatial tasks[24,63]. Our data, however, suggest that, during goal-directed behavior, the human hippocampus does not solely reflect the current state during navigation, or only the immediate future, but rather that it emphasizes strategically important states for reinstatement during ongoing behavior. Our results align with computational models that show that place cells associated with behaviorally relevant locations in an environment are preferentially incorporated into replay events[37].

### Relevance to models of hippocampal state space representation

Several models of hippocampal contributions to spatial navigation and memory propose that the hippocampus generates predictions of upcoming states[64]. For instance, a specific computational implementation of a predictive map model, the successor representation, states that the hippocampus is involved in learning relationships between states and actions, and that its representations reflect expectations about future locations[9,65]. We used a standard version of this computational model to generate simulated pattern similarity results, and surprisingly, these simulated matrices were qualitatively different from what we observed in the hippocampus.

In our simulations (see Supplemental Materials), a classical version of the successor representation reflected the transition probabilities between states, such that adjacent states were more similar than non-adjacent states. Because participants transitioned between all start and end positions equally in both directions, the model could not reproduce the difference between converging and diverging sequences either during the planning or navigation phases. It is possible that, in the relatively small and deterministic state space used in our task, it is not advantageous to represent an elaborate transition structure. An alternative approach to account for the present results would be to use a model that places heavier emphasis on context instead of only the next item or next decision. One model, the clone-structured cognitive graph model[23], is able to learn clones of similar observations that are distinguished by the current context. We predict that those models take into account context and goals, like the model presented in George et al., will be better able to capture the nuances of our task.

Alternatively, it might be advantageous to focus on models that incorporate an inductive bias to specifically focus on the most goal-relevant aspects of state space (e.g., the goal, context, and decision at P3). In many situations, an agent with an appropriate understanding of task structure could benefit by deploying the hippocampus more strategically, by preferentially encoding and prospectively retrieving

memories for key decision points towards a goal[12]. One example of a computational model that relies on strategic deployment of past experience comes from Lu, Hasson, and Norman[66]. Their simulations showed that it was computationally advantageous to prioritize hippocampal encoding and retrieval of temporally extended events at event boundaries, which are moments of high uncertainty or prediction error. In our task, inductive biases carried out through such a computational framework could emphasize the goal and key decision point, rather than passively representing all possible state transitions.

We hypothesize that hippocampal representations of physical space and abstract state spaces are flexible, reflecting the computational demands of the planning problem, and the participant's understanding of, and experience with, the problem[52,67]. In the present study, the task might have encouraged a model-based planning strategy in which future goals and key states are strategically retrieved and represented in hippocampus. In cases where learning is passive and incidental to the task, or when transitions between states change unpredictably, hippocampal state spaces might instead resemble successor-based maps. Finally, in more complex tasks, participants might adopt different strategies with varying degrees of emphasis on goal-relevant information[68].

Human behavior is characterized by the ability to plan and flexibly navigate decision spaces in order to realize future goals. The present findings suggest that the hippocampus represents context-specific, goal-oriented representations during navigation. These findings may contribute to the development of unified models accounting for hippocampal contributions to memory, navigation, and goal-directed sequential decision-making[69–71]. Additionally, this work highlights the importance of studying goal-directed behavior, attentional modulation of memory representations, and their impact on planning.

## Methods
### Participants
Thirty healthy English-speaking individuals participated in the fMRI study. All participants had normal or corrected-to-normal vision. Written informed consent was obtained from each participant before the experiment, and the Institutional Review Board at the University of California, Davis approved the study. Participants were compensated with an Amazon gift card for their time. Data from one participant was excluded due to technical complications with the fMRI scanner, one participant was excluded due to a stimulus computer malfunction, two participants were excluded due to poor behavioral performance in the scanner (defined as falling below-trained criterion, 85% correct, in the scanner), and one participant was removed from the scanner before the experiment concluded because they did not wish to continue in the study. Prior to data analysis, to ensure data quality, we conducted a univariate analysis to examine motor and visual activation during the task compared to an implicit baseline (unmodeled timepoints when the participant was viewing a fixation cross). Two participants showed little to no activation in these regions and were excluded from further analysis. The remaining 23 participants (11 male, 12 female, all right handed) are reported here.

### Stimuli and procedure
Data was collected from participants using Matlab 2016a and Psychophysics toolbox. Task stimuli consisted of nine common animals, shown in color on a grey background. Participants were tasked with learning two zoo contexts, consisting of animals organized in a specific spatial orientation (Fig. 1a). Importantly, animals in both contexts were visually identical, but each context had a distinct spatial organization. Training consisted of three stages per context: 1) map study, 2) exploration, 3) sequence navigation. This was followed by an additional sequence navigation phase that alternated between contexts.

During map study, participants were initially shown an overhead view of one of the zoo contexts (counterbalanced order across

participants). After studying this picture, participants were asked to reconstruct the location of all the animals by arranging icons on the screen. If participants were not able to perfectly recreate the maze they were shown the picture once more and asked to try again. Next, during the zoo exploration, participants used arrow keys to move between items in the zoo, starting from the central animal. At the bottom of the screen, participants were shown arrows indicating all possible moves from their current location (e.g. Left, Up, Down, Right at the center position of a maze). If participants made an incorrect move (moving outside of the animal maze) they were informed they made a wrong move. Participants were required to visit each of the animals four times before moving on to the next phase. During the sequence navigation phase, participants were shown a cue with a start and goal animal, and had four moves to reach the goal on a given trial. Start and goal animals were always the endpoints of an arm. Participants were trained to 85% criterion before learning the other context. The same training procedure outlined above was repeated for the second zoo context. After learning each of the zoos to criterion, participants completed an additional sequence navigation phase with the same timing as the MRI scanning session.

In the MRI scanner, participants completed six runs of the sequence navigation task (Fig. 1b). In each run, participants completed 16 sequence navigation trials. Trials from a given context were presented in a blocked fashion so that there were 8 consecutive trials from each context. Across runs, context blocks were alternated and their order was counterbalanced across participants. In addition, the order of sequences within each context was counter-balanced across blocks to ensure no systematic ordering effects influenced our results. Each navigation trial began with a cue signaling a start and a goal animal displayed for 3 s, followed by a 3 s ITI. Participants then saw the start animal and navigated by pressing buttons to move through the space one animal at a time. Animal items were displayed on the screen for 2 s with a 3 s ITI, regardless of a participant button press. For items where participants made a navigational error, text was displayed for 2 s informing them they made a wrong move or incorrectly navigated to a goal animal. In each zoo context, participants planned and navigated 12 distinct sequences (each repeated 4 times across 6 runs of scanning)

## MRI data acquisition
MRI data were acquired on a 3 T Siemens Skyra MRI using a 32-channel head coil. Anatomical images were collected using a T1-weighted magnetization prepared rapid acquisition gradient echo (MP-RAGE) pulse sequence image (FOV = 256 mm; TR = 1800 ms; TE = 2.96 ms; image matrix = 256 × 256; 208 axial slices; voxel size = 1 mm isotropic). Functional images were collected with a multi-band gradient echo planar imaging sequence (TR = 1222 ms; TE = 24 ms; flip angle = 67 degrees; matrix = 64 × 64, FOV = 192 mm; multi-band factor = 2; 3 mm³ isotropic spatial resolution).

## MRI data processing
Data were preprocessed using SPM12 (https://www.fil.ion.ucl.ac.uk/spm/) and ART Repair. Slice timing correction was performed as implemented in SPM12. We used the iterative SPM12 functional-image realignment to estimate movement parameters (3 for translation and 3 for rotation). Motion correction was conducted by aligning the first image of each run to the first run of the first session. Then all images within a session were aligned to the first image in a run. No participant exceeded 3 mm frame-wise displacement. A spike detection algorithm was implemented to identify volumes with fast motion using ART repair (0.5 mm threshold)[72]. These spike events were later used as nuisance variables within generalized linear models (GLMs). Participants native structural images were coregistered to the EPIs after motion correction. The structural images were bias corrected and segmented into gray matter, white matter, and CSF as implemented in SPM12. Native brainmasks were created by combining gray, white

matter masks. Data were smoothed with a 4 mm³ FWHM 3D gaussian kernel.

## Regions of interest
ROI definitions were generated using a combination of Freesurfer, and a multistudy group template of the medial temporal lobe. The multistudy group template was used to generate probabilistic maps of hippocampal head, body, and tail as defined by Yushkevich et al.[73] and warped to MNI space using Diffeomorphic Anatomical Registration Using Exponentiated Lie Algebra (DARTEL) in SPM8. Maps were created by taking the average of 55 manually-segmented ROIs and therefore reflect the likelihood that a given voxel was labeled in a participant. Masks were created by thresholding the maps at 0.5, (i.e., that voxel was labeled in 50% of participants). These maps were then reverse normalized to native space using Advanced Normalization Tools (ANTS). Participant-specific cortical ROIs were generated using Freesurfer version 6.0. from the Destrieux and Desikan atlas[74–76]. Individual cortical ROIs were binarized and aligned to participants' native space by applying the affine transformation parameters obtained during coregistration. These masks were combined into merged masks that encompassed the entire hippocampus bilaterally (see cue period pattern similarity for more information). Anatomical ROIs for V1/V2 and BA4a/p were obtained by running all participants structural scans through the freesurfer recon-all pipeline. Our V1/V2 ROI was obtained by merging the anatomical masks for BA17 and BA18 (Supplemental Fig. 2).

## Cue period pattern similarity analysis
Our primary interest was to investigate how prospective sequence representations were modulated based on context membership. To achieve this, we used representational similarity analysis to analyze multi-voxel activity patterns within regions of interest[77]. Generalized Linear Models (GLMs) were used to obtain single trial parameter estimates of the cue period using a modified least-squares all (LSA) model[35,78]. Data were high-pass filtered using a 128 s cutoff and pre-whitened using AR(1) in SPM. All events were convolved with a canonical HRF to be consistent with prior work[78]. Cue periods were modeled using separate single trial regressors for each cue (2 s boxcar). The remaining portions of the task were modelled as follows: Navigation periods were modelled with separate 25 s boxcar functions for each trial, separate single trial regressors for catch sequences modelled as a 15 s boxcar, separate single trial catch blank trials (stick function), outcome correct at condition level (stick), outcome incorrect at condition level (stick), and the four button presses at the condition level (stick). Nuisance regressors for motion spikes, 12 motion regressors (6 for realignment and 6 for the derivatives of each of the realignment parameters) and a drift term were included in the GLM. Pattern similarity between the resulting beta images were calculated using Pearson's correlation coefficient between all pairs of trials in the experiment. Only between run trial pairs were included in the analysis to avoid spurious correlations driven by auto-correlated noise[79].

Based on evidence of functional differentiation along the long-axis of the hippocampus, we tested for any longitudinal or hemispheric differences in hippocampal patterns[63,80,81]. Analyses revealed no significant differences in the pattern of results between left and right or between anterior or posterior segments of the hippocampus. As a result, subsequent analyses were performed with pattern similarity data from a bilateral hippocampus mask.

## Linear mixed models
Behavioral responses and pattern similarity were analyzed using linear mixed effects models to account for the nested structure of the dataset, allowing us to statistically model errors in our model clustered around individuals and trial types that violate the assumptions of

standard multiple regression models. Statistical comparisons were conducted in R (3.6.0) (https://www.r-project.org/) using lme4 and AFEX[82,83]. Reaction times were analyzed using the following formula:

$$(Figure1) : RT \sim Position + (1|participant) \qquad (1)$$

Where (1|participant) indicates the random intercept for participant and RT is the reaction time for each position during the navigation phase, excluding position 5 (as no response is required). Furthermore, outlier RTs were excluded that exceeded 2.5 standard deviations from a participant's average reaction time.

For the pattern similarity analyses, pairwise PS values were input for each participant into three separate models with the following formulas:

$$(Figure\ 2b) : PS \sim same\_sequence*same\_context + (1|participant) \qquad (2)$$

$$(Figure\ 2c/d) : PS \sim overlap*same\_context + (1|participant) \qquad (3)$$

$$(Supplemental\ Figure\ 2) : PS \sim move*same\_context + (1|participant) \qquad (4)$$

Where (1|participant) indicates the random intercept for participant and PS is the Pearson correlation coefficient for a given trial pair. Fixed effects for Eq. 2: (1) same sequence - a categorical variable with two levels indicating if the trial pair was from the same or different sequence. (2) Same context - categorical variable with two levels: same or different. Fixed effects for Eq. 3: overlap - a categorical variable with four levels: full, converging, diverging, and diff. start diff. goal. Same context - same as Eq. 2. Fixed effects for Eq. 4: Move - a categorical variable with three levels: same moves, shared moves, no moves. Same context - same as Eq. 2. Statistical significance for fixed effects was calculated by using likelihood ratio tests, a non-parametric statistical test where a full model is compared to a null model with the effect of interest removed. For example, to test the significance of an interaction term two models would be fit. One with two main effects and no interaction and the other with the interaction term. Effects sizes were calculated with the partial eta squared statistic. Follow up tests and estimated marginal means[84] from LMMs were calculated using the R package emmeans (https://cran.rproject.org/web/packages/emmeans/index.html). Effects sizes were calculated with Cohen's *d*.

In all the above models, a model with a maximal random effects structure, as recommended by Barr et al.[85], was first fit. In all cases the maximal model failed to converge or was singular, indicating over-fitting of the data. When examining the random effects structure for these models, random slopes for our fixed effects accounted for very little variance when compared to our random intercept for participant. To improve our sensitivity and avoid over-fitting these terms were removed as suggested by Matuschek et al[86]. Lastly, it is important to note that our results are not dependent on using linear mixed models. Using standard repeated measures ANOVA produces qualitatively and quantitatively similar results in all ROIs (See also Supplemental Tables 1–4).

## Successor representation simulation

To better understand specific predictions of the successor representation in our task we performed a simple simulation with respect to our task[9]. First, we created a topological structure (connected graph) that was similar to our task. As seen in Supplemental Fig. 1, this structure closely resembled the plus maze participants navigated in. We simulated the successor representation based on a random walk

policy using the equation.

$$M = (I - \gamma T)^{-1} \qquad (5)$$

Where $\gamma$ is a free parameter that controls the decay of the SR and T is the full transition matrix of the task depicted in Supplemental Fig. 1A/B. For the current simulations, gamma of 0.3 was used, but results are qualitatively similar for different values. Random walk or policy independence can be assumed in this case because maps were well learned before the scanner and each sequence was traversed in both directions an equal number of times[65].

We then tested the hypothesis that, during planning, the hippocampus encodes the SR of the first position in the sequence (columns of SR). We extracted columns of the SR for three planned sequences ((state 1 -> state 5) (state 6 -> state 5) (state 1 -> 9)) and calculated the similarity (Pearson's) between pairs of trials. The same sequence was calculated by correlating the same sequence with itself. The converging condition was obtained by correlating trials that started at different states but converged on the same end state. The diverging condition was obtained by correlating trials that started at the same state but diverged to different end state. Lastly, the diff. start diff. goal condition was calculated by correlating trial pairs that started and ended at different states. As shown in Supplemental Fig. 1, the SR heavily weights the immediate locations around the starting location and thus would predict that diverging sequences should have higher similarity than converging sequences.

## Timepoint-by-timepoint representational similarity analysis

To examine whether participants activated remote timepoints as they navigated through our virtual environments (e.g., activating decision points early in the navigation trial), we used a variant of single trial modeling using finite impulse response (FIR) functions[41]. This method allowed us to isolate the unique spatiotemporal pattern of activity for a given navigation trial while simultaneously controlling for surrounding time points during the run. We modeled 47 seconds of neural activity with a set of 38 FIR basis functions. Specifically, we obtained a spatial pattern of activity for each of these 38 TRs in our model, which allowed us to compare the similarity of the spatial patterns of activity between timepoints in the navigation phase. Additional regressors were included for motion, however spike regressors were not included in this analysis because they perfectly colinear with an FIR basis sets for each TR. A separate GLM was used for every trial resulting in 72 voxel time series. Collinearity in our model was measured using the variance inflation factor (VIF) and was verified to be within acceptable levels according to standards in the literature[87] (see also Supplemental Fig. 8). To examine within trial type similarity (same trial type across repetitions) timepoint-by-timepoint similarity matrices were generated by correlating activity patterns from repetitions of specific sequence pairs (e.g. zebra-tiger repetition 1 with camel-tiger repetition 1), at every TR. The resultant matrices were symmetrized by averaging across the diagonal of the matrix using the following equation: $(X^T + X)/2$. The resultant timepoint-by-timepoint similarity matrix was averaged within a specific trial type to get a single average timepoint-by-timepoint similarity matrix for each participant and condition (Fig. 3, Supplemental Fig. 7). This was done separately for converging and diverging sequences. Only between-run trial pairs were included in the analysis to avoid spurious correlations driven by auto-correlated noise[79]. This method allowed us to isolate individual sequence patterns while controlling for temporally adjacent navigation trials. To identify which points in time corresponded to relevant parts of the task, we manually lagged trial labels by 4 TRs to account for the slow speed of the HRF.

Time point-by-timepoint similarity matrices were constructed only for converging and diverging sequences. This subset of trials was chosen for several methodological reasons listed below. One is that, to

maximally control for differences in trial numbers between conditions and temporally auto-correlated evoked patterns, while still maintaining enough power to examine future state activation; we restricted our analyses to converging and diverging sequences within the same context. Importantly, this selection of trials allows us to simultaneously control for several factors while testing specific predictions. Another is that, converging and diverging sequences are matched in terms of the number of shared items and therefore overall visual similarity. Specifically, the same animal items are seen during the first half of diverging sequences, while the same animal items are seen in the second half of converging sequences (all sequences share the center item).

To assess statistical significance, and to correct for multiple comparisons, we used cluster-based permutation tests[88] with 10,000 permutations, with a cluster-defining threshold of 0.05 (two-tailed). Each pixel of a statistical comparison (T-value) was converted into a Z value by normalizing it to the mean and standard error generated from our permutation distributions. Cluster significance was determined by comparing the empirical cluster size to the distribution of the maximum cluster size (sum of T-values) across permutations with a cluster mass threshold of 0.05 (two-tailed).

### Reporting summary
Further information on research design is available in the Nature Portfolio Reporting Summary linked to this article.

## Data availability
Processed data to reproduce figures in the manuscript and supplement are available at https://github.com/jecd/Hippocampgoal and in a Zenodo repository https://doi.org/10.5281/zenodo.7264243[89]. Source data are provided with this paper. Raw data available at https://osf.io/txauh/.

## Code availability
Code to reproduce all figures and statistical analyses in the manuscript and supplement are available at https://github.com/jecd/Hippocampgoal and in a Zenodo repository at https://doi.org/10.5281/zenodo.7264243[89].

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

## Acknowledgements

We thank Charles Lowell for his assistance with data collection. We thank Trevor Baer, Costin Tanase, Dennis Thompson, and the Imaging Research Center for their technical contributions. We thank Matthias Gruber, Zach Reagh, Halle Dimsdale-Zucker, Nichole Bouffard, Walter Reilly and the Dynamic Memory Lab for consultation on analysis and experimental design. We thank James Antony for feedback on the manuscript. This research was funded by a Royal Society and Wellcome Trust Sir Henry Dale Fellowship to AC (211200/Z/18/Z), a Multi-University Research Initiative grant N00014-17-1-2961 from the U.S. Office of Naval Research/Department of Defense awarded to C.R., and grant N00014-20-1-2578 from the U.S. Office of Naval Research/Department of Defense awarded to Randall O'Reilly and C.R. Any opinions, findings, conclusions, or recommendations expressed in this material are those of the authors and do not necessarily reflect the official views of the Office of Naval Research, U.S. Department of Defense. For the purpose of open access, the author has applied a CC BY public copyright license to any Author Accepted Manuscript version arising from this submission.

## Author contributions

Conceptualization, J.C.D., A.C., CR.; Methodology, J.C.D., A.C., D.H., S.A.P., CR.; Investigation, J.C.D., A.C.; Writing – Original Draft, J.C.D., A.C., C.R; Writing – Review & Editing, J.C.D., A.C., S.A.P., D.H., E.D.B., C.R.; Funding Acquisition, C.R.; Supervision, C.R.

## Competing interests

The authors declare no competing interests.
