## [Peer Review File · Nature Communications]

Goal-oriented representations in the human hippocampus during planning and navigationREVIEWER COMMENTS

Reviewer #1 (Remarks to the Author):

This study aims to investigate how the hippocampus represents goals and contextual information during navigation. Participants completed a novel task in which they had to navigate around a 'zoo' while undergoing fMRI. Prior to entering the scanner, participants learned the layout of two distinct zoo 'contexts.' In each context, nine animals were arranged in a cross. In the scanner, participants completed 6 runs of the navigation task; each run included 8 blocked trials within a particular zoo 'context' followed by 8 trials in the other context. On every trial, participants saw a cue that depicted their starting animal location and animal location goal; they then used buttons to navigate from the starting location to the goal, viewing the animals they 'passed through' on their way. The researchers examined hippocampal pattern similarity within and between cue presentation on same-sequence, 'converging' (same goal), 'diverging' (same starting point trials, and no-overlap trials, both within and across contexts. Within a context, both same-sequence cues and converging cues were more similar to other same-sequence cues and converging cues, respectively, than they were to diverging or no-overlap cues. In addition, the researchers examined timepoint-by-timepoint hippocampal pattern similarity within converging and diverging sequences over the course of the trial. They found that pattern similarity between the first and third time point was higher for converging relative to diverging sequences. The researchers interpret these findings as indicating that the hippocampus represents contextual information about future states during goal-directed navigation, prioritizing goal- or sub-goal-relevant information.

Overall, this study uses a cleverly designed behavioral experiment to address an interesting question about hippocampal representations during goal-directed planning and navigation. However, I am not sure that the analyses presented here substantively advance our understanding of this topic, or that the authors' conclusions are fully justified based on the analyses and results described.

Broadly, the main finding of this study seems to be that at the start of sequential decisions, the hippocampus represents both contextual information and goal states. The notion that the hippocampus carries such information is not particularly novel. Indeed the authors review a number of prior findings demonstrating both goal encoding and contextual representations in the hippocampus. Moreover, in this task, both the goal state and the context are explicitly cued prior to each trial, which makes the fact that the hippocampus encodes this information especially unsurprising. While analyses are conducted that demonstrate representational similarity for shared destination trials than shared initial path trials (i.e., converging versus diverging trials), neither converging or diverging trial types exhibited significantly different pattern similarity than trials that had no path overlap, which calls into question the robustness of the effect.

The authors also conduct timepoint-by-timepoint analyses in which they test for evidence of prospective representation of future states, and find evidence of greater pattern similarity between the first and third time point for converging relative to diverging sequences. They interpret the results from these analyses as suggesting that for converging sequences, at State 1, participants represent State 3 (the center of the 'maze' in which they must make a turn toward the final goal). However, it is unclear why this relation should be stronger for converging vs. diverging sequences, since both groups of sequences share State 3. One control analysis that might strengthen this interpretation would be if the authors compared State 1 pattern similarity for all sequences that share the same State 3 and all those that do not. If pattern similarity at State 1 is higher for sequences that share the same State 3, this would strengthen the interpretation that the hippocampus represents the relevant subgoal at the first state.

In sum, I'm not sure that the findings presented in this paper afford a greater understanding of the role of the hippocampus in planning or navigation.

Below are a few suggestions for additional analyses or revisions:

Given the strong effects of context in the first set of hippocampal pattern similarity analyses, it seems as though the authors should also break down their time-point analyses by context. If the hippocampus is indeed carrying contextual information, then the expected differences in converging vs. diverging sequences may be obscured by this added 'noise.'

The introduction section was not particularly focused. It read more like a review paper than an exposition of the background and motivation for a specific empirical research question.

The introduction of the successor representation predictions and analyses in the discussion feels somewhat out of place.

Minor points:

Were trials in which participants made navigation mistakes excluded from neural analyses?

The authors note they used likelihood ratio tests to examine the significance of the fixed effects within their mixed-effects models, which only included random subject intercepts. This combination of intercept-only models and likelihood ratio tests is the least conservative way to test the significance of fixed effects (highest Type I error rates). Though the authors note the maximal models did not converge, they could further try:

- First removing the correlations between random slopes and intercepts
- Removing interaction terms in the random effects
- Removing just the random slope for the fixed effect that explains the least variance
- Using different optimizers

In addition, the authors could test the significance of their fixed effects via F tests.

Reviewer #2 (Remarks to the Author):

The Authors present a very nicely written manuscript investigating the involvement of the hippocampus during sequence navigation. They find that both planning and active movement through decision space modulate hippocampal activity in a context-dependent way, and show evidence that supports an alternative explanation of hippocampal coding to current goal states and those predicted by the successor representation. They find the hippocampal activity is highly context and goal dependent, and they support this with some very elegant control analysis. I found the study and analyses to be sound, but I would like to request some clarification on some of the results.

The comments are in chronological order (as manuscript was read):

In the text it says: $p_3 > p_4$, $z = 2.536$, $p = 0.0112$, but Figure 1 shows $p=0.055$ for that comparison, please indicate which it is.

The caption of Figure 2D says 'same context', is it meant to be 'different'? Relatedly, in the text (page 10 bottom) it seems that only the stats for panel C (within context) are reported. I was wondering what it means that between contexts, there is a significant effect of same sequence and converging sequences - I assume this means that the hippocampus is representing future goal states, but it is also context modulated (significant interaction). Are the values in the between context analysis significantly different from those in the within context analysis? Where could this context signal come from - is it possible to look at prefrontal areas to establish whether there is a context signal there driving the hippocampus? (especially as the task was blocked for context)

In caption S2: ' $\chi^2(2, N = 23) = 40.40$, $p < 0.0001$ ' is repeated and the word 'interaction' is in the

wrong place? (before p value). I really like these control analyses.

I find this section (page 16) a little confusing, because it reads as if the predictions would be early similarity for both convergent and divergent sequences (and the divergent show no such effect anyway). Also, was the statistical map also looked for the opposite subtraction, ie. D-C?)

"In this case, we would expect to see higher off-diagonal pattern similarity in converging sequences, such that across converging sequences, activity patterns associated with goal states would be correlated with activity patterns during earlier positions in the sequences. This effect should be higher for converging sequences than for diverging sequences because activations of state-action pairs should become more similar later in the sequence. Conversely, diverging sequence state-action pairs should be more similar earlier in the sequence but then decrease as the sequence progresses."

Relatedly, both from the timing of the clusters and the timecourse in Figure 4, it seems that it was actually position 4, rather than 5, that was reactivated, if at all (given this didn't survive MCC). However, I assume that after the decision point (P3), there was no need to necessarily plan ahead, as from then onwards, the same button presses need to be made. So to me, given this design, it makes sense that P3 is the most important (sub)goal, although I would then expect it in both convergent and divergent routes (if anything more in the latter, where different plans must be made at P3). In light of this, I do not think this statement in the Discussion is accurate: 'Similarly, during navigation, we found that the hippocampus prospectively activated information about upcoming states and that this effect was strongest in relation to key decision points and goals.' As goals were not significantly activated (p1 to p5 or p3 to p5).

Additionally, while the single subject maps are appreciated, it does seem that participant 19 is perhaps driving the group effect p1 to p3, as in the other subject plots, by eyeball, there isn't any clear evidence of this, if anything p5 seems to be most activated. Also, are the corrected deltaPS values in Figure S4 significant for either V1 or BA? as they also show p1 to p3 effects.

Comment (without having any personal stake in the validity of SR or any other theory): is it possible that with short and/or 'linear' routes, like in this task, there is no need to represent an elaborate transition structure? Perhaps if the sequence was longer or involved multiple choice points, the convergent/divergent effects would change. Relatedly, previous studies on hippocampal coding of distance often show correlations with differing directions - more/less hippocampal activity closer/further from the goal. My assumption, at least in these studies (as in Fig 3 of review by Spiers & Barry 2015 COBS, or even Figure 2 Patai 2019 Cerebral Cortex) is that there is an effect of environmental complexity / path complexity that contributes to these disparate findings. It is possible that the structure of the decision or state or physical space will have profound effects as to how the hippocampus is coding the relevant goal information.

For the TR analysis, were there any effects of the repetition of sequences across runs? Did earlier vs later repetitions show differing effects on the converging/diverging sequence analysis? I understand the sequences were very well learned, but it would be curious to know how repeating sequences affects the hippocampal representation. Relatedly, I assume the answer is yes, but just to check, were the 8 within context sequences presented in a block also counterbalanced? That is, it didn't always start with zebra to tiger etc. And equally, there wasn't any grouping of routes that were subsequently defined as convergent/divergent?

Reviewer #3 (Remarks to the Author):

The authors present a very interesting and timely study of how goal-oriented representations in the human hippocampus manifest in spatially-grounded sequence processing. In particular, they tackle the question of how overlap affects the expression of these codes, and their results suggest that 1)

hippocampal representations of future states emerge in a context-bound manner and 2) their expression is modulated in a dynamic way by how the memory crosses paths with others.

I was generally very enthusiastic about this work, and the methods employed (which were largely clearly presented as well), and the bigger questions it addresses in the literature. I did have several comments and concerns on interpretational and methodological points that I believe could strengthen the manuscript further when addressed.

1. I think when the authors argue for a positive modulation of similarity by goal that can't be explained by modulation by shared motor plans, they could test for a significant interaction between the two goal and move-set dimensions to formally demonstrate that the modulation is more strongly driven by the goal representation in a specific context than the move sequence. However, I would also add that I'm not sure a null interaction (should that be the case) would detract much from the manuscript. This is because although move sequences are motoric, they also correspond to location and heading shifts, which are reflected in the behavior of some neurons of the hippocampal system in rodents.

2. This reasoning led me to a related comment: did the authors consider testing a more hierarchical coding perspective? Akin to McKenzie ...Eichenbaum, 2014, it seems highly likely that even if there is a relative dominance of goal coding, it could be quite informative to think about how this is modulated but "subordinate" representations of other dimensions of the task.

3. Were timecourses time-shifted for interpretation of when representational information emerged? That is, when the authors attribute a TR pattern to time position 3, is that adjusted for lag in the hemodynamic response that state (say, taking ~TR 3 after position 3)?

4. A related comment is that the authors describe these (very interesting!) timecourse outcomes as "the first item in the sequence activating the central position" -> yet directionality cannot be assessed from this alone. This is interesting to consider, because it is noteworthy that it is for the diverging routes, more so than the converging routes, that there is an alternative decision about future state 3 that could be prospectively made at time 1. At least, computing this sequence element in advance may be more behaviorally-beneficial than look-ahead to a choice that simply leads to the same stimulus regardless of route (the converging scenario). Could their pattern similarity outcome instead reflect a representation of the alternative starting path at the point in the environment where they intersect (that is, when arriving at state 3, a representation of the other memory leading to this state is elicited)? In theory, I do think the high p1-p1 similarities for converging routes speak against this prior notion somewhat, however (but please see next comment).

5. Yet, it is also troubling that in both the converging and diverging cases the pattern similarity is high early (e.g., p1-p1) but declines uniformly forward in time even on the diagonal (also relevant to my query about time-shifting in the analysis). This also results in little evidence of the goal or "goal arm sequence" itself being instantiated earlier (i.e., p1-end similarity). This result from the timecourse analysis suggests the main trial-level analysis results using LSA may be tracking an early abstracted state in the trial more than a representation that is present at the goal itself. I found this difficult to interpret, even though I agreed with the authors' statements to the effect that the data seemed to be about an abstract goal more than specific stimuli.

This fading in similarity later in the trial, even in a converging arm case where the goal and items could be the same) did leave me wondering if there is a chance there could be an impact of the TR-by-TR FIR modeling approach. Although the FIR predictors are not fit sequentially (that I can tell) – could later points in the time window be more driven by noise than signal, perhaps due to an artifact of some structure in the collinearity of the FIR hrf predictors? Some deeper discussion of this pattern in the results, and perhaps potential impact of modeling could strengthen the arguments made.

Minor:

In the methods, the authors describe the timeseries analysis as yielding "72 voxel timeseries" but I believe this is a typo

I thought it could be worth a Discussion sentence or two juxtaposing the current study results more directly with the Chanales... Kuhl 2017 paper, which the authors did cite. Despite some differing outcomes, there are a lot of structural similarities in the environment, and a small comment on this may prompt formal comparative research ideas.

REVIEWER COMMENTS (in black) and Responses (in blue)

We appreciate the reviewers' enthusiasm for our work and constructive comments about how we could improve the manuscript in a revision. We have addressed all of the reviewers' comments below and we have incorporated these changes into our manuscript, as indicated below.

Reviewer #1 (Remarks to the Author):

This study aims to investigate how the hippocampus represents goals and contextual information during navigation. Participants completed a novel task in which they had to navigate around a 'zoo' while undergoing fMRI. Prior to entering the scanner, participants learned the layout of two distinct zoo 'contexts.' In each context, nine animals were arranged in a cross. In the scanner, participants completed 6 runs of the navigation task; each run included 8 blocked trials within a particular zoo 'context' followed by 8 trials in the other context. On every trial, participants saw a cue that depicted their starting animal location and animal location goal; they then used buttons to navigate from the starting location to the goal, viewing the animals they 'passed through' on their way. The researchers examined hippocampal pattern similarity within and between cue presentation on same-sequence, 'converging' (same goal), 'diverging' (same starting point trials, and no-overlap trials, both within and across contexts. Within a context, both same-sequence cues and converging cues were more similar to other same-sequence cues and converging cues, respectively, than they were to diverging or no-overlap cues. In addition, the researchers examined timepoint-by-timepoint hippocampal pattern similarity within converging and diverging sequences over the course of the trial. They found that pattern similarity between the first and third time point was higher for converging relative to diverging sequences. The researchers interpret these findings as indicating that the hippocampus represents contextual information about future states during goal-directed navigation, prioritizing goal- or sub-goal-relevant information.

Overall, this study uses a cleverly designed behavioral experiment to address an interesting question about hippocampal representations during goal-directed planning and navigation. However, I am not sure that the analyses presented here substantively advance our understanding of this topic, or that the authors' conclusions are fully justified based on the analyses and results described.

- 1) Broadly, the main finding of this study seems to be that at the start of sequential decisions, the hippocampus represents both contextual information and goal states. The notion that the hippocampus carries such information is not particularly novel. Indeed the authors review a number of prior findings demonstrating both goal encoding and contextual representations in the hippocampus. Moreover, in this task, both the goal state and the context are explicitly cued prior to each trial, which makes the fact that the hippocampus encodes this information especially unsurprising. While analyses are conducted that demonstrate representational similarity for shared destination trials than shared initial path trials (i.e., converging versus diverging trials), neither converging or diverging trial types exhibited significantly different pattern similarity than trials that had no path overlap, which calls into question the robustness of the effect.

Response #1

We thank Reviewer 1 for raising these points and appreciate the opportunity to demonstrate both the novelty and robustness of our experimental findings.

We now realize that the introduction of our original submission did not clearly convey the major disconnect in the literature. There is now a large body of neuroimaging work on representation of abstract state spaces in the hippocampus, and this work has been guided by theories that rely heavily on studies of hippocampal place cells during random foraging. Tolman (1948), however, proposed the concept of the cognitive map to explain how mammals can flexibly plan routes in order to actively navigate towards a goal. Based on what is known about hippocampal activity during planning or goal-directed spatial navigation, there is reason to believe that planning and navigation in abstract spaces might emphasize goals, rather than representations of current or possible states, as suggested by prominent influential models of hippocampus (e.g. O’Keefe and Nadel, 1978, Stachenfeld et al., 2017). Thus, our study addressed a fundamental gap in the literature.

Another key variable is context. Studies of abstract state space representation by the hippocampus have investigated mapping of passively- or incidentally- learned relationships between stimuli, and to our knowledge, no prior studies have investigated whether such representations are context-dependent. We know, however, that hippocampal place cells show a high degree of context-specificity, such that it constructs different maps for different contexts. If hippocampal maps of abstract spaces are context-specific, that would place fundamental limits on current models, that do not explain learning and representations of contexts.

Here, we investigated planning and goal-directed navigation in two different state-space contexts—each involving the same stimuli, but with different relationships between the stimuli. As shown in the paper, our findings diverge considerably from current “predictive map” models which suggest that the hippocampus learns distributions of probable future states.

We have thoroughly rewritten the introduction so that the gap in knowledge that our study addressed is more apparent to the reader. In addition, to clarify the novelty and significance of our results, we have added the following sections to the results section to more accurately depict current theories on the role of the hippocampus during planning and navigation. We hope that this will address Reviewer 1’s concerns.

Added to Introduction Pg. 3 – 5:

“ Several lines of evidence suggest that the hippocampus plays a key role in navigation, though its role in navigation is fundamentally unclear. For example, based on findings showing that hippocampal “place cells” encode specific locations within a spatial context, many have argued that the hippocampus forms a cognitive map of physical space (O’Keefe and Dotrovsky, 1971; O’Keefe and Nadel, 1978). It is now clear that the hippocampus also tracks distances in abstract state spaces (Tavares et al., 2015; Park et al., 2019; Aronov et al., 2017), potentially supporting

the broader idea that the hippocampus encodes a “memory space” (Eichenbaum & Cohen, 2014) that maps the systematic relationships between any behaviorally relevant variables (Behrens et al 2018, Stachenfeld et al., 2017, Kaplan, Schuck, & Doeller 2017; but see O’Reilly et al. 2022 and Summerfield et al., 2020 for alternative views).

Building on this idea, some have proposed that the hippocampus encodes a “predictive map” that specifies not only one’s *current* location, but also states or locations that could be encountered in the future (e.g. Mehta et al., 2001, Stachenfeld et al., 2017). For example, the “successor representation,” a popular computational implementation of the predictive map model (e.g. Gershman, 2018), has been used to argue that the hippocampus represents each state in terms of its possible transitions to future states. This model demonstrates that through an incremental learning process about state-to-state transitions, analogous to model-free learning about rewards, enables organisms to rapidly learn how a sequence of actions can lead to a desired outcome.

Although numerous studies have investigated representations of abstract state spaces in the hippocampus, two fundamental questions remain unanswered. One key issue concerns the role of context. Single-unit recording studies have reported that the spatial selectivity of place cells is context-specific—that is, the spatial selectivity of a given cell in one environment varies when an animal is moved to a different, but topographically similar environment (O’Keefe and Dostrovsky, 1971, Skaggs and McNaughton, 1998; Leutgeb et al., 2004, Alme et al., 2014, McKenzie et al., 2014). Just as one might pull up different cognitive maps for different physical contexts, it is reasonable to think that we might utilize context-specific maps of abstract state spaces. Computational models have been proposed to explain how the hippocampus might recognize contexts (Honi et al., 2020, Whittington et al., 2020, George et al., 2021), but there is little empirical evidence showing whether or how context is utilized in abstract spaces.

A second key issue that has not been addressed, concerns how goals affect hippocampal representations of abstract task states. Theories of state space representation by the hippocampus rely heavily on results from studies that examined activity in hippocampal place cells during random movements through an environment (e.g. Alme et al., 2014). Accordingly, studies of abstract spaces in humans typically investigate incidental learning of stimulus dimensions or arbitrary state dynamics (Garvert et al., 2017, Schapiro et al 2016, Schuck & Niv, 2016). These kinds of passive, incidental learning tasks differ from those used by Tolman (1948) to demonstrate that animals actively use a spatial representation to guide navigation to particular goal locations in an environment. If the human hippocampus forms an abstract cognitive or predictive map, one would expect to see such a representation during planning and navigation towards different goals in the same context.

Based on what is known from studies of spatial navigation, there is reason to think that hippocampal representations in the context of goal-directed navigation might fundamentally differ from what is seen during random or incidental behavior. For example, hippocampal place cells have differential firing fields during planning depending on the future goal of the animal (Ainge et al., 2007; Wood et al., 2000; Ferbinteanu and Shapiro, 2003, Ito et al., 2015), and goal locations tend to be overrepresented (Dupret et al., 2010, Gauthier et al., 2018). Consistent with these findings fMRI studies of spatial navigation have found that hippocampal activity is modulated by a participant’s distance from a goal location (Patai et al., 2019, Howard et al., 2014), and that hippocampal activity patterns during route planning carry information about prospective goal locations in a virtual space (Brown et al., 2016). These findings suggest that hippocampal representations during planning or navigation in abstract state spaces might be

powerfully shaped by goals. If this is indeed the case, it would potentially challenge models proposing that the hippocampus encodes a relatively static map of current (O'Keefe and Dotrovsky, 1971) or possible future states (Stachenfeld et al., 2017).”

Added to Representation of Behaviorally Relevant Sequence Positions During Navigation Pg. 14
-17:

“Having established that the hippocampus represents information about context-specific goals during planning, our next analyses turned to how state-action information is dynamically represented during navigation. Available evidence suggests at least three ways that navigationally-relevant information might be represented by the hippocampus. Based on classic studies of place cells, we might expect the hippocampus to represent the current state as participants navigated toward the goal. Alternatively, based on predictive map models (Stachenfeld et al., 2017), we could expect that the hippocampus would represent not only the current state but also future states.

A third possibility is that the hippocampus might preferentially represent goal-relevant information during navigation. In our study, the most behaviorally significant points in a navigated sequence were the starting point (position 1), when a goal-directed plan must be initiated, and the center of the maze (position 3), a critical sub-goal where one’s decision will determine the ultimate trial outcome. This was confirmed by our behavioral analyses that revealed that participants were slower to respond at positions 1 and 3 (**Fig. 1**). We therefore reasoned that participants might be likely to prospectively retrieve hippocampal representations of these states during navigation.

To test this prediction, we examined pattern similarity differences during navigation across converging and diverging sequences in the same zoo context. Converging and diverging sequences were chosen because these sequences have an equal number of overlapping states, but the timing of the overlap is systematically different. Both the “current state” and standard “predictive map” models would suggest that pattern similarity during navigation should reflect this pure overlap--early in a sequence there should be higher pattern similarity across pairs of diverging sequence trials, and late in a sequence there should be higher pattern similarity across pairs of converging sequence trials. In contrast, a goal-based account would predict that pattern similarity could reflect prospective coding of goal-relevant information (e.g. He., et al 2022, Brown et al., 2016) which should be higher across converging sequences (which share the same upcoming goal), relative to diverging sequences (which overlap in early states but lead to different goals).

We used a time-point by time-point pattern similarity analysis approach that enabled us to examine information in multivoxel activity patterns about current, past, and future states to test our key hypotheses. This technique is conceptually similar to cross-temporal generalization techniques used in pattern classification analyses (King & Dehaene, 2014). First, we extracted the time-series for each navigation sequence using a variant of single trial modeling that utilizes finite impulse response (FIR) functions (Turner et al., 2012), allowing us to examine activity patterns for each time point (TR) as participants navigated through the sequence of items. Importantly, incorrect trials were excluded from this analysis. As depicted in **Figure 3**, we quantified pattern similarity between pairs of navigation sequences (e.g. zebra to tiger sequence compared to camel to tiger sequence) at different timepoints (e.g., TR 1 to TR 10), which yielded a timepoint-by-timepoint similarity matrix for each condition (converging or diverging sequences). The diagonal elements for this matrix reflect similarity between pairs of animal

items from the same timepoint in the sequence. Off-diagonal elements reflect the similarity between an animal at one timepoint in the sequence and animal items at other timepoints in the sequence.

Separate timepoint-by-timepoint correlation matrices (Pearson's r) were created for pairs of converging sequence trials and pairs of diverging sequence trials. We next computed a difference matrix and tested for statistically significant differences between converging and diverging sequences, correcting for multiple comparisons using cluster-based permutation tests (10,000 permutations, see Methods for more details).

As noted above, diverging sequences have overlapping states early in the sequence, and converging sequences have overlapping states late in the sequence. If the hippocampus represents only current states, we would expect to see pattern similarity differences between converging and diverging close to the diagonal of the timepoint-by-timepoint matrices — that is, we would expect higher pattern similarity for diverging pairs during timepoints early in the sequence and higher pattern similarity for converging pairs during timepoints late in the sequence. If the hippocampus represents current and temporally-contiguous states, as suggested by predictive map models, we would expect that at early positions, we would expect higher pattern similarity for diverging sequences, both on- and off-diagonal, and at late positions, we would expect higher pattern similarity for converging sequences both on- and off-diagonal. Finally, if the hippocampus preferentially represents goal-relevant information during navigation (Mattar et al., 2018, He et al., 2022), we would expect to see higher off-diagonal pattern similarity only for *converging* sequences, because only converging sequences share the same goal. Specifically, we expected higher off-diagonal pattern similarity between goal states and earlier positions in the sequences.”

In addition, goal context is not explicitly cued on a per trial basis. Instead, the context is cued in a blocked fashion such that at the beginning of a scanning run a specific zoo context is cued. Then, the following 8 navigation trials for one context are performed before receiving a cue for another context where the next 8 navigation trials are performed.

It is not clear why the use of explicit cues may be seen as problematic, as nearly all studies of goal-directed navigation in rodents and humans use some form of explicit cuing or instruction. Our task is designed in such a way that if participants are not cued with a specific goal for a given trial, then participants would not be able to form a plan during the cue period or be able to navigate to their goal. These behaviors are central to our empirical questions. How the hippocampus functions to retrieve the correct context-specific memory, in the face of many highly overlapping paths, is not known. As we now clarify in the introduction of the paper (Pg. 4), a number of tasks have investigated passive or incidental learning of task state relationships, and these tasks are very different from the kind of goal-directed navigation tasks used by Tolman, and in subsequent studies of hippocampal function by Wood and Eichenbaum, Shapiro, and others.

It is conceivable that the use of visual cues could introduce some effect of visual similarity that could influence pattern similarity results. To demonstrate that visual similarity did not impact our results, we looked specifically at differences between Converging and Diverging sequences in relation to the Same Sequence. If the hippocampus only cared about the amount of visual overlap (e.g. sharing start or sharing goal), we should see no difference between Converging and

Diverging cues. We have conducted several other control analyses outlined in the results and supplement (Figure S2) that specifically rule out the role of visual or motor confounds in the effects observed in the hippocampus.

It might be useful to note that we were surprised at the results. In our revised manuscript, we have emphasized how our results deviate from current theories of hippocampal state-space representation. A simple episodic memory (Davachi, 2006) or episodic simulation (Schacter et al., 2007) hypothesis would have predicted that pattern similarity should simply reflect the overlap in states on a route, such that pattern similarity should be highest for identical route trials, intermediate for same start or same goal trials, and lowest for different start/different goal trials. As we have detailed in our simulations using the successor representation, our results were also unexpected based on what might be predicted from the “predictive map” hypothesis. As we now clarify in the discussion section, we think that the best way to explain the present results is that, in the context of goal-directed behavior, hippocampal memory functions might be strategically deployed to emphasize decision-relevant states, as opposed to mapping all relationships, in order to help construct a model-based plan.

Reviewer 1’s concerns also alerted us to the fact that our use of the term “no overlap” was misleading. In fact, in our study, every route involved overlap, as the third state was identical in all routes. This is analogous to studies of spatial navigation with “plus maze” topologies, in which an animal must return to the center of the maze on every journey. It is also important to note that trial pairs that were originally labeled as “no overlap” included pairs of trials with identical stimuli but in reversed order. For example, this could include cues from the zebra to tiger sequence and the tiger to zebra sequence (same exact stimuli but in a reverse order and mirrored moves). Thus, the condition labeled as “no overlap” included trials with a number of overlapping states. To rectify this issue, we have relabeled the no overlap condition to “Different Start/Different Goal” to better reflect this. We also highlight this point in the paper in the results and methods sections, as well as any figures where we illustrate the trial types that are included in each of our conditions.

Finally, it is important to emphasize that the robustness of the results is especially evident when considering results across contexts. Looking across contexts, converging sequences are significantly different from diff. start/diff. goal. Interestingly, the directionality here is that converging sequences show significantly higher similarity than diff. start/diff. goal across contexts while diverging sequences have lower pattern similarity than diff. start/diff. goal, though not significant. (converging > diff. start/diff. goal across context: $z = 2.08$, $p = 0.038$; diverging > diff. start/diff. goal across context: $z = 1.02$, $p = 0.31$). Taken together, we interpret these results to suggest that both goal and context are highly relevant for activity patterns in the hippocampus during planning. Importantly, when looking across sequences that share a goal, context has a differential effect on sequential plans, as indicated by a significant interaction between goal and context.

- 2) The authors also conduct timepoint-by-timepoint analyses in which they test for evidence of prospective representation of future states, and find evidence of greater pattern similarity between the first and third time point for converging relative to diverging sequences. They interpret the results from these analyses as suggesting that for

converging sequences, at State 1, participants represent State 3 (the center of the 'maze' in which they must make a turn toward the final goal). However, it is unclear why this relation should be stronger for converging vs. diverging sequences, since both groups of sequences share State 3. One control analysis that might strengthen this interpretation would be if the authors compared State 1 pattern similarity for all sequences that share the same State 3 and all those that do not. If pattern similarity at State 1 is higher for sequences that share the same State 3, this would strengthen the interpretation that the hippocampus represents the relevant subgoal at the first state.

In sum, I'm not sure that the findings presented in this paper afford a greater understanding of the role of the hippocampus in planning or navigation.

Below are a few suggestions for additional analyses or revisions:

Response #2

We appreciate this reviewer's thoughtful comment and we will attempt to clarify the theoretical logic behind this conclusion. Prospectively activating a sequence of steps requires you to activate memory representations associated with your upcoming plan. We use representational similarity to compare repetitions of different sequences that have different starting states but the same endpoint. We compare this similarity to sequences that have the same starting point but have different endpoints.

We acknowledge this reviewer's criticism here that both sequences share P3 and it is not clear why there should be any difference between converging and diverging sequences. Our perspective is that at P1 both converging and diverging sequences should be activating P3. However, based on our results we hypothesize that activity patterns evoked by P3 do not only contain P3 information. Rather, they also include pattern information related to their ultimate goal (P5) and also possibly other sensorimotor information important for realizing that goal. We think that this makes sense if one assumes that the *hippocampus is not representing the item at P3, but rather it is representing the information that is relevant to reaching the goal*. Note that P3 is the key point of uncertainty in the task, and at this position, the agent's decision should be dictated by the goal. For converging sequences P1-P3 similarity is driven by the fact that subjects are using the same goal to guide planning on the move at P3. In diverging sequences P1-P3 similarity is still driven by those same factors, but subjects' plans are dictated by different goals, and thus result in lower pattern similarity.

Lastly, we have attempted to clarify the predictions and importance of this result by more clearly laying out possible predictions in the manuscript. We have also clarified our interpretation by adding the following sentences to the discussion section. An important point to emphasize is that all sequences share State 3 (P3) and thus we cannot complete this control analysis as described in comment #2. We hope that our explanations to the reviewer and additional clarifications added to the manuscript address Reviewer #1's concerns.

The hippocampus represents context-specific goal information during planning Pg: 22

“In contrast, in our experiment, all trials that converged on the same goal required the same key decision at position 3, regardless of the starting point. In this situation, it is optimal to learn a representation that captures the information that is common to any sequence that converges on the same goal. For example, as depicted in Figure 1, any trial with a tiger as the goal animal will require participants to choose the “down” button at position 3. In the next section, we explain why results from the navigation period are also consistent with this interpretation.”

Reinstatement of remote timepoints in the hippocampus during navigation Pg. 23-24:

“As noted above, the animals in the first three positions overlapped across diverging sequences, whereas the animals in the last three positions overlapped across converging sequences. Thus, if the hippocampus only represented the current state during navigation, we would have expected pattern similarity on the diagonal in **Figure 4** to be higher for diverging trials for early time points, and then higher for converging trials in the later time points (see also Figure **S4**). Instead, we found that the significant differences between converging and diverging trial pairs were primarily off of the diagonal, suggesting that, during the navigation phase, hippocampal patterns carried information about behaviorally relevant remote timepoints along the route. More specifically, hippocampal activity patterns early in the navigation phase carried information about position 3 in converging trials, as compared to diverging trials.

This pattern of results is notable because the stimulus at position 3 is exactly the same for all trials in all contexts, so these results could not solely reflect prospective retrieval of future stimuli. As noted above, the correct decision to be made at position 3 depends on one’s current goal and context. All converging sequences, which share the same goal, require the same decision at P3, whereas diverging sequences are associated with different decisions at P3 because they involve different goals. These results are consistent with the idea that rather than carrying information about sequences of upcoming states, participants were prospectively activating the most goal-relevant information in the upcoming sequence, namely the context- and goal-appropriate decision at position 3.”

- 3) Given the strong effects of context in the first set of hippocampal pattern similarity analyses, it seems as though the authors should also break down their time-point analyses by context. If the hippocampus is indeed carrying contextual information, then the expected differences in converging vs. diverging sequences may be obscured by this added ‘noise.’

Response #3

We appreciate the reviewer’s comment and apologize if this point was not clear in our original manuscript. All time-point by time-point analyses were conducted on trials within the same context. We have attempted to clarify this point in our manuscript by adding the following sentences to our results section to be more explicit.

Representation of behaviorally relevant sequence positions during navigation - Pg, 15:

“To test this prediction, we examined pattern similarity differences during navigation across converging and diverging sequences in the same zoo context. Converging and diverging

sequences were chosen because these sequences have an equal number of overlapping states, but the timing of the overlap is systematically different.”

To further address this comment, we have also included additional analyses contrasting converging and diverging sequences in different contexts as well as testing for an interaction across contexts (Converging – Diverging Same Context > Converging – Diverging Different Context). This analysis was conducted by using the same analysis procedure as our other TR by TR analyses (See Figure 3, 4 and Methods). We have included below two additional visualizations of this result (Reviewer 1 Figure 1 and Figure 2).

Reviewer 1 Figure 1: Converging > Diverging Sequences in Different Contexts

A) - Group level pattern similarity results from converging sequences in different context during active navigation. **B)** - Same as A) but showing diverging sequences in different contexts. **C)** - TR by TR pattern similarity results depicting a statistical map of converging – diverging. Z values were calculated using a bootstrap shuffling procedure with 10,000 permutations. **D)** – Thresholded statistical map at $p < 0.025$. Cluster based permutation tests with 10,000 permutations (Maris and Oostenveld, 2007) were performed with a cluster defining threshold of $p < 0.025$ and a cluster alpha of 0.05. Note that no clusters survive multiple comparisons correction.

As can be seen above in Reviewer 1 Figure 1 there is no clear evidence of significant off diagonal activation for Converging vs. Diverging sequences across contexts. Given our interpretation of our results, this suggests that in converging sequences within the same context, people are activating upcoming representations of a critical decision point that is context specific. This is because across context subjects are activating different context specific plans and thus similarity across contexts should be low.

Reviewer 1 Figure 2: (Converging > Diverging Sequences Same Context) > Converging > Diverging Sequences Different Context

A) - Group level pattern similarity results from converging sequences > diverging sequences in the same context during active navigation. **B)** - Same as A) but showing diverging sequences. **C)** - TR by TR pattern similarity results depicting a statistical map of the interaction effect. Z values were calculated using a bootstrap shuffling procedure with 10,000 permutations. **D)** - Thresholded statistical map at $p < 0.025$. Cluster based permutation tests with 10,000 permutations (Maris and Oostenveld, 2007) were performed with a cluster defining threshold of $p < 0.025$ and a cluster alpha of 0.05. Note that no clusters survive multiple comparisons. Outlined in Red is the cluster extent of the cluster identified in Converging > Diverging Same Context.

We also tested conducted our TR by TR analyses on the interaction effect of converging and diverging sequences across contexts. As can be seen in Reviewer 1 Figure 2 no clusters survive multiple comparisons correction but interestingly, we still see higher pattern similarity around time points that would be associated with activating position 3 while at position 1. In addition, we also have highlighted the cluster extent of the Converging > Diverging Same Context to illustrate graphically that the effects observed occur in approximately the same remote timepoints.

Taken together we feel that these two data points illustrate the specificity of our effect to the same context and demonstrate visually that timepoints clustered around position 3 show the biggest effects when comparing across contexts. We hope that these address Reviewer 1's concerns with the robustness of our TR by TR similarity effects.

- 4) The introduction section was not particularly focused. It read more like a review paper than an exposition of the background and motivation for a specific empirical research question.

Response #4

We appreciate Reviewer 1's constructive feedback on ways we can improve the focus of the introduction. We have reworked the introduction of the paper to be more focused on the specific empirical questions addressed in the paper and the gaps in the literature that our study fills. Mainly, we have removed text from the intro that is not directly related to hippocampal activity in planning, navigation, and memory. The introduction now has a clear flow from Hippocampus -> Cognitive and Predictive Maps -> Context and Goals -> Our Experiment. Given the intersectional nature of the research conducted here, we also felt it was necessary to have a broad introduction spanning several different bodies of literature to accurately motivate the importance and novelty of our study. We think that this has improved the manuscript overall and hope that this will address Reviewer 1's concerns.

- 5) The introduction of the successor representation predictions and analyses in the discussion feels somewhat out of place.

Response #5

We appreciate this reviewer pointing out areas for us to improve the focus of our manuscript. As mentioned in Response #1 and Response #2 above, we have extensively reworked both the introduction and results section to help make the motivations for our successor representation analyses clearer. To be explicit the successor representation theory is now described broadly in our paper as the "Predictive Map Hypothesis". In the Discussion, we have added a subheading: "Relevance to models of hippocampal state space representation." Under this subheading, we provide context for our simulations using the successor representation, and consider alternative models that might more accurately capture the present results. Lastly, the primary purpose of the Successor Representation simulations was to formalize predictions from a standard predictive map model and to illustrate that our hippocampal results cannot be explained solely by

accounting for shared transition probabilities (which would predict higher similarity for Diverging Sequences).

Added to Discussion Pg. 25-26

Relevance to models of hippocampal state space representation

Several models of hippocampal contributions to spatial navigation and abstract state spaces propose that the hippocampus generates predictions of upcoming states. For instance, a specific computational implementation of a predictive map model, the successor representation, states that the hippocampus is involved in learning relationships between states and actions, and that its representations reflect expectations about future locations (Stachenfeld et al., 2017; Momennejad, 2020). We used a standard version of this computational model to generate simulated pattern similarity results, and surprisingly, these simulated matrices were qualitatively different from what we observed in the hippocampus.

In our simulations (see Supplemental Materials), a classical version of the successor representation reflected the transition probabilities between states, such that adjacent states were more similar than non-adjacent states. This is because, participants transitioned between all start and end positions equally in both directions. Thus, the model could not reproduce the difference between converging and diverging sequences either during the planning or navigation phases. It is possible that in the relatively small and deterministic state space used in our task, it is not advantageous to represent an elaborate transition structure. In larger decision spaces, our data would suggest that during navigation the hippocampus would be involved in activating memories for key decision points towards a goal. An alternative approach to account for the present results would be to use a model that places heavier emphasis on context instead of only the next item or next decision. One model, the “clone-structured cognitive graph” model (George et al., 2021), is able to learn “clones” of similar observations that are distinguished by the current context. We predict that that models that take into account context and goals, like the model presented in George et al., will be better able to capture the nuances of our task.

Alternatively, it might be advantageous to focus on models that incorporates an inductive bias to specifically focus on the most goal-relevant aspects of a state space (e.g., the goal, context, and decision at P3). In this case, we would expect that a single algorithm such as the SR could account for all kinds of state space representations in the hippocampus. Instead, hippocampal representations of physical space (Ekstrom and Ranganath, 2017) and abstract state spaces (Boorman, Schweigert, & Park, 2021) are likely to be more flexible, reflecting the computational demands of the planning problem, the subject's experience with the problem, and the situation. In the present study, the task might have encouraged a model-based planning strategy, in which future goals and key states are strategically retrieved and represented in hippocampus. In other tasks, where the structure is well learned and people need to re-plan, hippocampal state spaces might resemble successor-based maps.

Minor points:

- 6) Were trials in which participants made navigation mistakes excluded from neural analyses?

Response #6

We thank this reviewer for mentioning this point and we have included the following text in the manuscript to address this concern and clarify which trials were included or excluded from different analyses.

Results Cue Period Pg. 8-9:

“In addition, only trials which resulted in subjects subsequently making the correct moves towards the goal were included in neural analyses.”

Results Nav Period Pg. 16:

“Importantly, incorrect trials were excluded from this analysis”

- 7) The authors note they used likelihood ratio tests to examine the significance of the fixed effects within their mixed-effects models, which only included random subject intercepts. This combination of intercept-only models and likelihood ratio tests is the least conservative way to test the significance of fixed effects (highest Type I error rates). Though the authors note the maximal models did not converge, they could further try:
 - First removing the correlations between random slopes and intercepts
 - Removing interaction terms in the random effects
 - Removing just the random slope for the fixed effect that explains the least variance
 - Using different optimizers

In addition, the authors could test the significance of their fixed effects via F tests.

Response #7

We appreciate the helpful comments for how to properly control for type 1 error rates using mixed effects models. As described in the methods under the linear mixed effects model section, we followed the procedure that is described in Matuschek, 2017. This procedure first involves fitting the maximal model as described in Bates et al., 2013. This resulted in the model failing to converge or producing a singular model fit (which indicated overfitting). We then removed the correlation between random slopes and intercepts and refit the model. Then we removed the interaction terms in the random effects and refit the model, checking for convergence. Lastly, we removed the random slopes for fixed effects that explained the least variance (items). This procedure was followed for all mixed effect models used in the paper and resulted in the final model reported in the methods section of the paper (all resulted with only random intercepts for subjects).

We did not try different optimizers, but feel as though we have been sufficiently rigorous in following the established guidelines for using and reporting results obtained from mixed effects models.

In our revised manuscript, we demonstrate the robustness of our effects by showing that the selection of statistical tests does not impact our main experimental findings. Per the reviewers' suggestions, we have provided tables for the two main models used within bilateral hippocampus with both likelihood ratio tests and F-tests. As can be seen below, the Sequence by Context

interaction and Overlap by Context analyses are statistically significant regardless of the type of statistical test used. In summary these analyses increase our confidence in the robustness of our results. This table has been added to the supplemental materials.

Table 1 Sequence By Context Likelihood Ratio

Table

DF full model: 6

Effect	DF	Chi Sq.	P Val.
Sequence	1	3.57	0.059
Context	1	3.36	0.067
Sequence * Context	1	4.26	0.039

Table 2 Sequence By Context F-Tests Table

Effect	DF	F	P Val.
Sequence	1,66	3.5	0.066
Context	1,66	3.3	0.074
Sequence * Context	1,66	4.2	0.044

Table 3 Overlap Analysis Likelihood Ratio Table

DF full model: 10

Effect	DF	Chi Sq	P Val.
Context	1	2.03	0.15
Overlap	3	4.85	0.18
Context * Overlap	3	14.75	0.002

Table 4 Overlap Analysis F-Tests Table

Effect	DF	F	P Val.
Context	1,154	1.95	0.16
Overlap	3,154	1.57	0.2
Context * Overlap	3,154	4.93	0.003

We have added to the results section a note to the reader guiding them to the methods where they can find detailed information about model selection for linear mixed effects models. Detailed discussions of the model selection approach are too verbose for the main body of the paper, but we point this reviewer to the methods section where we clearly outline the procedure followed in Matuschek, 2017.

Methods - Linear Mixed Model Section, Pg. 35:

‘In all the above models, a model with a maximal random effects structure, as recommended by Barr et al., 2014, was first fit. In all cases the maximal model failed to converge or was singular indicating over-fitting of the data. When examining the random effects structure for these

models, random slopes for our fixed effects accounted for very little variance when compared to our random intercept for subject. To improve our sensitivity and avoid over-fitting these terms were removed as suggested by Matuschek et al., 2017. Lastly, it is important to note that our results are not dependent on using linear mixed models. Using a standard repeated measures ANOVA produces qualitatively and quantitatively similar results in all ROIs.”

Reviewer #2 (Remarks to the Author):

The Authors present a very nicely written manuscript investigating the involvement of the hippocampus during sequence navigation. They find that both planning and active movement through decision space modulate hippocampal activity in a context-dependent way, and show evidence that supports an alternative explanation of hippocampal coding to current goal states and those predicted by the successor representation. They find the hippocampal activity is highly context and goal dependent, and they support this with some very elegant control analysis. I found the study and analyses to be sound, but I would like to request some clarification on some of the results.

The comments are in chronological order (as manuscript was read):

We appreciate the reviewer’s enthusiasm about our manuscript. We also thank the reviewer for their careful read of our manuscript and for their helpful suggestions. We have addressed the reviewer’s comments below and we have incorporated these changes into our manuscript, as we indicate below.

The comments are in chronological order (as manuscript was read):

- 1) In the text it says: $p_3 > p_4$, $z = 2.536$, $p = 0.0112$, but Figure 1 shows $p=0.055$ for that comparison, please indicate which it is.

Response #1

We appreciate the reviewer catching this typographic error. The statistics reported in the text are correct, and this is a mistake in the figure. To be clear for the p_3 vs p_4 comparison: “ $p_3 > p_4$, $z = 2.536$, $p = 0.0112$ ”. We have updated the figure to correctly reflect the statistics reported in the paper.

- 2) The caption of Figure 2D says 'same context', is it meant to be 'different'? Relatedly, in the text (page 10 bottom) it seems that only the stats for panel C (within context) are reported. I was wondering what it means that between contexts, there is a significant effect of same sequence and converging sequences - I assume this means that the hippocampus is representing future goal states, but it is also context modulated (significant interaction). Are the values in the between context analysis significantly different from those in the within context analysis? Where could this context signal come from - is it possible to look at prefrontal areas to establish whether there is a context signal there driving the hippocampus? (especially as the task was blocked for context)

Response #2

We thank Reviewer 2 for these thoughtful observations. Figure 2D is attempting to illustrate the directionality of the context by overlap interaction. Values greater than 0 in this plot indicate that pattern similarity for a given condition is higher for the within context comparison. Values below 0 indicate higher similarity for a given condition for a between context comparisons. Both Same and Converging sequences show significantly higher pattern similarity when in the same context (same: $z = 2.60$, $p = 0.0094$; converging: $z = 2.51$, $p = 0.012$). This is in contrast to diverging sequences which show a trending pattern in the opposite direction ($z = -1.89$, $p = 0.060$); higher pattern similarity for sequences that come from different contexts. Related to this point, we appreciate the suggestion to compare the pattern similarity values from figure 2C and 2D. However, Figure 2D is already doing exactly this. Put another way, we are already comparing the sequence representation for each of the conditions in Figure 2C across contexts in Figure 2D.

We have added the following text to the results section to properly report the statistical comparisons conducted in this plot. We also added a guide to the axis for figure 2D to indicate what values greater than and less than zero signify.

Added to results section page 12:

“Between contexts, cues of the same sequence and converging sequences showed significantly higher pattern similarity when in the same context (Same Sequence: $z = 2.60$, $p = 0.0094$; Converging: $z = 2.51$, $p = 0.012$). In contrast, diverging sequences showed a different pattern of results such that sequences from different contexts had higher similarity ($z = 1.89$, $p = 0.060$). Lastly, sequences with different starting states and goals were not significantly modulated by context ($z = 0.430$, $p = 0.67$).”

Our interpretation of this result is that across contexts, specifically when you are activating a similar plan, the hippocampus represents these events in a similar way if they are in the same context. This is a standard context dependent memory effect that has been reported in the Hippocampus and MTL many times before. What is novel here is that even different sequences (Converging) that share the same goal show a surprising effect when considering standard models of the hippocampus’ role in memory. See Reviewer 1 Response #1 and #2 for a longer discussion of why this result is novel and the empirical motivations for our study.

Lastly, regarding the role of prefrontal areas in our task, it is highly likely that the regions in the PFC are contributing to the pattern of results observed in the hippocampus. Unfortunately, when we examined different sub regions of the PFC (mainly vmPFC) we did not observe significant effects for any of our primary analyses. Specifically, when conducting our primary overlap by context analysis on a vmPFC ROI obtained from the Desitrieux atlas in Freesurfer (Fischl et al., 2004, Desitrieux et al., 2010). We find no significant effects of context or overlap during planning see table and figure below. It would be worthwhile to further investigate the role of the PFC in future studies looking at how goal information impacts hippocampal representations.

Table 5 Overlap By Context Likelihood Ratio Table

DF full model: 10

Effect	DF	Chi Sq.	P Val.
Context	1	0.05	0.82
Overlap	3	6.5	0.09
Context * Overlap	3	0.54	0.91

- 3) In caption S2: ' $\chi^2(2, N = 23) = 40.40, p < 0.0001$ ' is repeated and the word 'interaction' is in the wrong place? (before p value). I really like these control analyses.

Response #3

We thank this reviewer for pointing this typographic error out and have updated the manuscript supplement accordingly.

- 4) I find this section (page 16) a little confusing, because it reads as if the predictions would be early similarity for both convergent and divergent sequences (and the divergent show no such effect anyway). Also, was the statistical map also looked for the opposite subtraction, ie. D-C?) "In this case, we would expect to see higher off-diagonal pattern similarity in converging sequences, such that across converging sequences, activity patterns associated with goal states would be correlated with activity patterns during earlier positions in the sequences. This effect should be higher for converging sequences than for diverging sequences because activations of state-action pairs should become more similar later in the sequence. Conversely, diverging sequence state-action pairs should be more similar earlier in the sequence but then decrease as the sequence progresses."

Response #4

This section, pages 14-16, are meant to lay out possible outcomes for how this analysis may have turned out, which is supported by past literature on the topic. To clarify, we had three hypotheses for differences in pattern similarity in converging and diverging sequences: (1) The hippocampus represents the current state (2) The hippocampus represents the past and future state in addition to the current state (3) The hippocampus preferentially represents goal information during navigation.

We have re-worked this section (pages 14-16) so that it more clearly lays out our predictions and the possible outcomes of this analysis. See Reviewer 1 Response #1 above for the full text addition.

We appreciate the reviewer's interest in understanding the directionality of the time-point by time-point effect. We used cluster-based permutation tests (Maris & Oostenveld, 2007) using a two-sided t-test with a cluster defining threshold of 0.05. This means that to initially define clusters to be included in our permutation test, we examined both positive and negative clusters. Put another way, we examined both the Converging - Diverging and the Diverging - Converging statistical maps. To add additional clarification, during the permutation test step, separate monte carlo simulations were conducted for both positive and negative clusters to determine their significance at a threshold of $p < 0.05$. These values are standard in the literature and is what is suggested on the Fieldtrip website.

- 5) Relatedly, both from the timing of the clusters and the timecourse in Figure 4, it seems that it was actually position 4, rather than 5, that was reactivated, if at all (given this didn't survive MCC). However, I assume that after the decision point (P3), there was no need to necessarily plan ahead, as from then onwards, the same button presses need to be made. So to me, given this design, it makes sense that P3 is the most important (sub)goal, although I would then expect it in both convergent and divergent routes (if anything more in the latter, where different plans must be made at P3). In light of this, I do not think this statement in the Discussion is accurate: 'Similarly, during navigation, we found that the hippocampus prospectively activated information about upcoming states and that this effect was strongest in relation to key decision points and goals.' As goals were not significantly activated (p1 to p5 or p3 to p5).

Response #5

We thank Reviewer 2 for this question about which specific timepoints are being activated. First, we agree with the reviewer's point that it is not appropriate to make strong claims about results in the timecourse analysis that do not survive multiple comparisons correction. Accordingly, we have scaled back claims about prospective activation of goal locations and focused instead on the prospective activation of the decision point sub-goal. As shown in Figure 4E, pattern similarity is significantly higher for a cluster of time points that correspond roughly to increased similarity between P1 and P3. This finding survives multiple comparisons correction, and therefore we now have focused our conclusions on this finding.

Regarding the reviewer's second point, we acknowledge that the same stimulus is shown at P3 on every trial. If one were to prospectively activate the *item* shown at P3 on every trial, we would expect no significant differences between converging and diverging sequences. However, our data suggests that, at the early stages of navigation, the hippocampus prospectively represents the decision point (P3) in a goal-directed manner, such that the same position is represented differently according to the ultimate destination.

We think that this makes sense if one assumes that the *hippocampus is not representing the item at P3, but rather it is representing the information that is relevant to reaching the goal*. Note that P3 is the key point of uncertainty in the task, and at this position, the agent's decision should be dictated by the goal. For converging sequences P1-P3 similarity is driven by the fact that subjects are using the same goal to guide planning on the move at P3. In diverging sequences P1-P3 similarity is still driven by those same factors, but subjects' plans are dictated by different goals, and thus result in lower pattern similarity. We have now clarified this interpretation of the results in the discussion section of the revised manuscript. See also Reviewer 1 Responses #1 and #2.

The hippocampus represents context-specific goal information during planning Pg: 22

“In contrast, in our experiment, all trials that converged on the same goal required the same key decision at position 3, regardless of the starting point. In this situation, it is optimal to learn a representation that captures the information that is common to any sequence that converges on the same goal. For example, as depicted in Figure 1, any trial with a tiger as the goal animal will require participants to choose the “down” button at position 3. In the next section, we explain why results from the navigation period are also consistent with this interpretation.”

Reinstatement of remote timepoints in the hippocampus during navigation Pg. 23-24:

“As noted above, the animals in the first three positions overlapped across diverging sequences, whereas the animals in the last three positions overlapped across converging sequences. Thus, if the hippocampus only represented the current state during navigation, we would have expected pattern similarity on the diagonal in **Figure 4** to be higher for diverging trials for early time points, and then higher for converging trials in the later time points (see also Figure **S4**). Instead, we found that the significant differences between converging and diverging trial pairs were primarily off of the diagonal, suggesting that, during the navigation phase, hippocampal patterns carried information about behaviorally relevant remote timepoints along the route. More specifically, hippocampal activity patterns early in the navigation phase carried information about position 3 in converging trials, as compared to diverging trials.

This pattern of results is notable because the stimulus at position 3 is exactly the same for all trials in all contexts, so these results could not solely reflect prospective retrieval of future stimuli. As noted above, the correct decision to be made at position 3 depends on one's current goal and context. All converging sequences, which share the same goal, require the same decision at P3, whereas diverging sequences are associated with different decisions at P3 because they involve different goals. These results are consistent with the idea that rather than carrying information about sequences of upcoming states, participants were prospectively activating the

most goal-relevant information in the upcoming sequence, namely the context- and goal-appropriate decision at position 3.”

- 6) Additionally, while the single subject maps are appreciated, it does seem that participant 19 is perhaps driving the group effect p1 to p3, as in the other subject plots, by eyeball, there isn't any clear evidence of this, if anything p5 seems to be most activated. Also, are the corrected deltaPS values in Figure S4 significant for either V1 or BA? as they also show p1 to p3 effects.

Response #6

We are happy to provide additional visualizations to demonstrate that our effects are not driven by outliers. Below is a plot where we have calculated the average PS values from the timepoints identified using our cluster-based permutation tests from each subject. You can clearly see in this plot that subject 19 and also subject 15 do indeed show a larger effect in those timepoints. However, 13 other subjects also show modest effects during those same time points suggesting that this effect is consistent across participants. In other words, *20 of 23 subjects showed effects in the same direction*—higher p1 to p3 pattern similarity for converging than for diverging sequence pairs. Formally, this was confirmed by conducting a Wilcoxon signed rank test on the single subject similarity values from the figure below. This confirmed that the majority of participants showed an increase in pattern similarity within these timepoints (Signed Rank = 238, $Z = 3.042$, $p = 0.0024$). Furthermore, we have chosen a stringent statistical test to correct for multiple comparisons and utilized data driven monte carlo simulations to generate our null distribution. These methods for assessing statistical significance are robust to outliers.

The reviewers also asked whether the corrected deltaPS values in Figure S4 were significant for visual or motor cortex. In fact, in visual (cluster mass: 348.2348, $p = 0.0052$) and motor (cluster mass: 673.7853, $p < 0.0001$) cortex, P1-P3 pattern similarity was higher for converging than diverging sequences. These findings are consistent with the overall interpretation that, early in the trial, participants prospectively activate a representation of p3, which is the decision point.

We believe that the increased off-diagonal pattern similarity effects in visual and motor cortex reflects prospective activation of the relevant sensory and motor representations at p3. In other words, participants likely are predicting the animal at p3 and the button press that will be made at p3 in order to navigate to the goal. This type of predictive coding has been shown in primary sensory areas in past work and is a possible mechanism for the effects that we observe here (Hindy et al., 2016, Clarke et al., 2021).

We have adjusted the supplement figure to more closely mirror the hippocampal figure (outlined significant clusters in Red) in the manuscript. In addition, we have added associated p-values with the clusters to the supplemental figure caption. We hope that this addresses Reviewer #2's concerns and provide clarification on the interpretation of our results.

- 7) Comment (without having any personal stake in the validity of SR or any other theory): is it possible that with short and/or 'linear' routes, like in this task, there is no need to represent an elaborate transition structure? Perhaps if the sequence was longer or involved multiple choice points, the convergent/divergent effects would change. Relatedly, previous studies on hippocampal coding of distance often show correlations with differing directions - more/less hippocampal activity closer/further from the goal. My assumption, at least in these studies (as in Fig 3 of review by Spiers & Barry 2015 COBS, or even Figure 2 Patai 2019 Cerebral Cortex) is that there is an effect of environmental complexity / path complexity that contributes to these disparate findings. It is possible that the structure of the decision or state or physical space will have profound effects as to how the hippocampus is coding the relevant goal information.

Response #7

We thank Reviewer 2 for these insightful comments and questions. We agree with this reviewer that in larger decision spaces it is likely inefficient to represent an elaborate transition structure (See Baluger et al., 2016). Based on our hippocampal navigation data, we would predict that, instead of representing an elaborate transition structure, the hippocampus would preferentially represent actions taken at key decision points, in relation to the subjects' current goals. This prediction could not be tested in the present paradigm, but we plan to test it in future work involving navigation in more complex contexts with multiple choice points. In addition, we agree that that the structure of the task or state space will have profound impacts on goal representation in the hippocampus and in interconnected cortical networks (see Ekstrom & Ranganath, 2018).

Regarding distance to goal coding, akin to Spiers and Barry 2015 and Patai 2019, we do find that univariate activity in the anterior hippocampus increases as participants traverse the sequence. These data were not included in the manuscript because, as Reviewer 2 states, these effects have been shown elsewhere.

Based on the reviewer's suggestion, we have added the following sentences to the discussion:

Discussion Pg. 25-26:

“It is possible that in the relatively small and deterministic state space used in our task, it is not advantageous to represent an elaborate transition structure. In larger decision spaces, our data would suggest that during navigation the hippocampus would be involved in activating memories for key decision points towards a goal. An alternative approach to account for the present results would be to use a model that places heavier emphasis on context instead of only the next item or next decision. One model, the “clone-structured cognitive graph” model (George et al., 2021), is able to learn “clones” of similar observations that are distinguished by the current context. We predict that that models that take into account context and goals, like the model presented in George et al., will be better able to capture the nuances of our task.”

- 8) For the TR analysis, were there any effects of the repetition of sequences across runs? Did earlier vs later repetitions show differing effects on the converging/diverging sequence analysis? I understand the sequences were very well learned, but it would be curious to know how repeating sequences affects the hippocampal representation. Relatedly, I assume the answer is yes, but just to check, were the 8 within context sequences presented in a block also counterbalanced? That is, it didn't always start with zebra to tiger etc. And equally, there wasn't any grouping of routes that were subsequently defined as convergent/divergent?

Response #8

We thank the reviewer for this comment. We apologize if the counterbalancing approach used in our experiment was unclear and have added text to the main results section. To clarify, sequences were counter balanced across blocks So that the order of the sequences was different in each block. In addition, each block had a unique combination of individual sequences that were presented. E.g. block 1 had seq 1, 2, 4, 5, 6, 7, 8. Block 2 had seq 4, 8, 9, 12 etc.

The following sentence was added to the methods section Pg. 30:

In addition, the order of sequences within each context was counter-balanced across blocks to ensure no systematic ordering effects influenced our results.

Related to the above point, the counterbalancing was done in a way to maximize our capabilities to do between run pattern similarity. Thus, sequence repetitions are not evenly spaced across blocks and it would be difficult to look at changes in PS values across runs in a balanced way. For example, the zebratiger sequence may be presented in context 1 in the first run. Then would not be presented again until the 4th, 5th, and 6th runs. In this example, this would force us to only examine pattern similarity for that sequence in late blocks.

In addition, all PS values presented in this manuscript are calculated between run in order to avoid temporal autocorrelation that may artificially inflate PS values (e.g. Hendrickson et al., 2015). As a result, the different repetitions (both early and late) contribute to PS values presented here and PS cannot be evaluated effectively in our design for a targeted repetition pair. It is an interesting empirical question to better understand how hippocampal representations evolve over the course of learning. We hope that future researchers can investigate this question.

Reviewer #3 (Remarks to the Author):

The authors present a very interesting and timely study of how goal-oriented representations in the human hippocampus manifest in spatially-grounded sequence processing. In particular, they tackle the question of how overlap affects the expression of these codes, and their results suggest that 1) hippocampal representations of future states emerge in a context-bound manner and 2) their expression is modulated in a dynamic way by how the memory crosses paths with others.

I was generally very enthusiastic about this work, and the methods employed (which were largely clearly presented as well), and the bigger questions it addresses in the literature. I did have several comments and concerns on interpretational and methodological points that I believe could strengthen the manuscript further when addressed.

We appreciate the reviewer's enthusiasm for our work and we appreciate their helpful comments. We have addressed their suggestions below and we have modified our manuscript to address these issues.

- 1) I think when the authors argue for a positive modulation of similarity by goal that can't be explained by modulation by shared motor plans, they could test for a significant interaction between the two goal and move-set dimensions to formally demonstrate that the modulation is more strongly driven by the goal representation in a specific context than the move sequence. However, I would also add that I'm not sure a null interaction (should that be the case) would detract much from the manuscript. This is because although move sequences are motoric, they also correspond to location and heading shifts, which are reflected in the behavior of some neurons of the hippocampal system in rodents.

Response #1

We appreciate this reviewer's suggestion to help us strengthen one of the main findings of the paper. However, we are not able to complete this analysis exactly as the reviewer suggested. In order to fit a model that has additional regressor for move-set it would require us to look across contexts. This is because within a context diverging and converging sequences are perfectly colinear with the move-set dimension. This is by design and allows us to control for visual similarity, but by doing so were not able to fully control for motor sequence similarity.

As this reviewer mentioned, we attempted to investigate the impact of motor overlap on the similarity structure observed in our data in Figure S2. We were able to demonstrate that similarity cannot be solely explained by motor overlap and this model shows no significant main effects nor significant interactions. To attempt to address this reviewer's concern, we have conducted one follow-up test on Same Moves Same Context (equivalent to Same Sequence Same Context) compared to Same Moves Different Context in our Bilateral Hippocampal ROI (See Figure S2B). This contrast specifically tests for the impact of moves on goal representations. If the goal representations we observed in the data were only driven by movement information we should see no difference between these two conditions. This test revealed a numerically different but non-significant difference between the conditions (Same Move Same Context > Same Move Different Context: $z = 1.68$, $p = 0.093$). As a positive control

we also ran this same contrast in our primary motor ROI. Here we would expect no modulation by context because this region should only be driven by motor information and not goal information. Indeed, we observe no significant difference between Same Move Same Context and Same Move Different Context ($z = 0.80, p = 0.43$).

We interpret these findings as the hippocampus being somewhat driven by move similarity but also by goal information. However, the null difference between Same Moves Same Context vs. Same Moves Different Context is especially difficult to interpret. Like this reviewer mentioned, this is due to the fact that the shared moves across contexts are not exactly equivalent. Moves across contexts are different in both their position and heading shifts. We have omitted this contrast from the manuscript and feel that our primary control analyses are sufficient to address this reviewers concern.

- 2) This reasoning led me to a related comment: did the authors consider testing a more hierarchical coding perspective? Akin to McKenzie ...Eichenbaum, 2014, it seems highly likely that even if there is a relative dominance of goal coding, it could be quite informative to think about how this is modulated but “subordinate” representations of other dimensions of the task.

Response #2

We appreciate this reviewers’ interest in our work and their suggestion to investigate more hierarchical representations that may be present in the data. In response to this suggestion, we have completed an analysis akin to Mckenzie et al., 2014 below.

We averaged pairwise correlation matrices within conditions (e.g. Zebra Tiger Cx. 1 Rep. 1, Rep 2. etc.) and across participants to create a group-level condition by condition similarity matrix that was sorted by context. As can be seen in panel A and B, we did not find clear evidence of a hierarchical representation in the group level matrices. To further visualize this effect and look for the possibility of a more hierarchical coding scheme (goals nested within contexts), we then constructed a dendrogram. This dendrogram was constructed using the unweighted average distance (Pearson’s) between our conditions. Interestingly, this revealed some notable examples

of sequences that share the same goal being part of neighboring leaves. However, it is not apparent from these visualizations alone that there are sub-clusters of sequences that share the same goal. This pattern of results neither refutes nor clearly supports the possibility that there is a hierarchical coding within the hippocampus. It is important to note that this analysis is done on average correlation matrices and doesn't account for individual differences in sequence representations. Moreover, one of our main findings (Figure 2D) is that there is a heterogeneous effect of context. Where sequences that share the same goal are grouped together within the same context while sequences that do not have the same goal are not as strongly impacted by context.

- 3) Were timecourses time-shifted for interpretation of when representational information emerged? That is, when the authors attribute a TR pattern to time position 3, is that adjusted for lag in the hemodynamic response that state (say, taking \sim TR 3 after position 3)?

Response #3

We thank the reviewer for this comment and apologize if this was unclear in the manuscript. To clarify, pattern similarity values in our TR by TR analyses were manually lagged by 4 TRs (TR = 1.22, Inter-Item-Interval = 5s) to account for the hemodynamic response lag. This information can be found in the methods under the TR by TR analysis section. We have added the following text to the figure 4 caption to help other readers better understand our analyses.

“Trial labels were manually lagged by 4 TRs (TR = 1.22, Inter-Item-Interval = 5s) to account for hemodynamic response lag.”

- 4) A related comment is that the authors describe these (very interesting!) timecourse outcomes as “the first item in the sequence activating the central position” -> yet directionality cannot be assessed from this alone. This is interesting to consider, because it is noteworthy that it is for the diverging routes, more so than the converging routes, that there is an alternative decision about future state 3 that could be prospectively made at time 1. At least, computing this sequence element in advance may be more behaviorally-beneficial than look-ahead to a choice that simply leads to the same stimulus regardless of route (the converging scenario). Could their pattern similarity outcome instead reflect a representation of the alternative starting path at the point in the environment where they intersect (that is, when arriving at state 3, a representation of the other memory leading to this state is elicited)? In theory, I do think the high p1-p1 similarities for converging routes speak against this prior notion somewhat, however (but please see next comment).

Response #4

We thank this reviewer for their enthusiasm for our results and their thoughtful comments. It is interesting to consider that in the converging condition the similarity we observe is driven by some common process that is evoked at P3 that reflects the differential paths that led to their current positions. Below we attempt to lay out our interpretation of these results.

Our perspective is that at P1 both converging and diverging sequences should be activating P3. However, based on our results we hypothesize that activity patterns evoked by P3 do not only contain P3 information. Rather, they also include pattern information related to their ultimate goal (P5) and also possibly other sensorimotor information important for realizing that goal. For converging sequences P1 P3 similarity is driven by the fact that subjects are all planning/predicting/moving to the next stimulus in a similar way across repetitions of that sequence. In diverging sequences P1 P3 similarity is still driven by those same factors, but subjects' plans/predictions/moves lead to divergent outcomes and thus result in lower pattern similarity.

We have added a few sentences to the discussion and results to clarify the above logic (See also Reviewer 2 Response #5 and Reviewer 1 Response #1 and #2). We hope that this addresses this Reviewer # 3's comment.

Reinstatement of remote timepoints in the hippocampus during navigation Pg. 23-24:

“As noted above, the animals in the first three positions overlapped across diverging sequences, whereas the animals in the last three positions overlapped across converging sequences. Thus, if the hippocampus only represented the current state during navigation, we would have expected pattern similarity on the diagonal in **Figure 4** to be higher for diverging trials for early time points, and then higher for converging trials in the later time points (see also Figure **S4**). Instead, we found that the significant differences between converging and diverging trial pairs were primarily off of the diagonal, suggesting that, during the navigation phase, hippocampal patterns carried information about behaviorally relevant remote timepoints along the route. More specifically, hippocampal activity patterns early in the navigation phase carried information about position 3 in converging trials, as compared to diverging trials.

This pattern of results is notable because the stimulus at position 3 is exactly the same for all trials in all contexts, so these results could not solely reflect prospective retrieval of future stimuli. As noted above, the correct decision to be made at position 3 depends on one's current goal and context. All converging sequences, which share the same goal, require the same decision at P3, whereas diverging sequences are associated with different decisions at P3 because they involve different goals. These results are consistent with the idea that rather than carrying information about sequences of upcoming states, participants were prospectively activating the most goal-relevant information in the upcoming sequence, namely the context- and goal-appropriate decision at position 3.”

- 5) Yet, it is also troubling that in both the converging and diverging cases the pattern similarity is high early (e.g., p1-p1) but declines uniformly forward in time even on the diagonal (also relevant to my query about time-shifting in the analysis). This also results in little evidence of the goal or “goal arm sequence” itself being instantiated earlier (i.e., p1-end similarity). This result from the time course analysis suggests the main trial-level analysis results using LSA may be tracking an early abstracted state in the trial more than a representation that is present at the goal itself. I found this difficult to interpret, even though I agreed with the authors' statements to the effect that the data seemed to be about an abstract goal more than specific stimuli.

This fading in similarity later in the trial, even in a converging arm case where the goal and items could be the same) did leave me wondering if there is a chance there could be an impact of the TR-by-TR FIR modeling approach. Although the FIR predictors are not fit sequentially (that I can tell) – could later points in the time window are more driven by noise than signal, perhaps due to an artifact of some structure in the collinearity of the FIR hrf predictors? Some deeper discussion of this pattern in the results, and perhaps potential impact of modeling could strengthen the arguments made.

Response #5

We understand Reviewer 3's point about the decline in pattern similarity over the course of sequence navigation, and appreciate the chance to clarify this issue. As we note in the introduction, previous fMRI studies have investigated representations of state spaces during incidental exposure to specific task states (e.g., during a task involving decisions about particular stimuli) or exposure to passively learned sequences with precise timing. To our knowledge, no prior fMRI studies have utilized *active*, self-initiated navigation through a state-space. The reason, we suspect, is that the timing of cognitive activities is less constrained during self-initiated navigation. At the beginning of a trial during the planning period, it is likely that all participants generated a mental model of their key moves in their plan to reach the goal for the trial, but the exact timing of processes related to prospection and planning during navigation is likely to vary significantly across individuals (we note that work on individual differences in proactive vs. retroactive control is relevant to this point; cf. Braver et al., JoCN, 2021). This, plus the considerable overlap of states within the diverging sequence pairs, makes it remarkable that any significant differences were seen in the time course analysis. We would be happy to add some discussion of this issue to the supplementary materials or the main body of the manuscript if Reviewer 3 believes that it would be helpful.

With regard to the second point concerning goal-arm coding, we now take a more conservative approach in our revised manuscript. Based on comments from this reviewer and from Reviewer 1, we no longer make conclusions about prospective activation of the goal representation during navigation. Instead, we focus on the finding that survived multiple comparisons correction, which is that there was significantly greater activation of P3 during converging trials than during diverging trials. We conclude that this reflects the fact that P3 is the critical position where one's decision is dictated by the current goal. Rather than activating the stimulus at P3 (which is common to all sequences), we believe participants are retrieving P3's relationship to the goal in the current zoo context. See also Reviewer 2 Response #5 and Reviewer 1 Response #2 for a more detailed discussion of the theoretical logic that supports this idea.

Reviewer 3 Figure 1 – Single subject design matrix and model diagnostics for FIR modelling approach. **Top Left Panel:** A single subject design matrix for one sequence modelled with our FIR approach. The orange, green and red bars highlight parameters associated with current sequence being modelled, nuisance regressors, and motion parameters respectively. **Top Right Panel:** A parameter by parameter collinearity matrix measured with cosine similarity. The lower left and diagonal elements have been excluded to aid in visualization. Values of 0 illustrate orthogonal regressors whereas values closer to 1 illustrate a degree of collinearity. The orange box outlines the rows of the matrix that are associated with the sequence being modeled. **Bottom Panel:** Variance inflation factor as a function of individual parameters. The orange bar is intended to direct the reviewer to timepoints that are used in RSA. Note that parameters associated with the sequence being modeled have relatively stable VIF and are within commonly held standards within the literature (Mumford et al., 2015).

Finally, the reviewer was concerned that that the results might have been systematically affected by the FIR modeling approach. The short answer is that there is no aspect of the modeling procedure that would systematically reduce pattern similarity values at later time points, relative to values for earlier timepoints. To clarify this issue, we have provided a deeper dive into the analysis procedure. The elements of the FIR basis set are, indeed, fitted simultaneously to the data, not in a stepwise fashion. We have provided an example single trial model matrix and corresponding model diagnostics to help illustrate the procedure.

As shown in the collinearity matrix, there was not a systematic difference in collinearity between FIR regressors modeling activity early- vs. late- in the trial. To measure the effect of collinearity in our model estimates we used a metric called variance inflation factor (VIF) (Belsley et al., 1980). This metric is calculated by predicting a held out regressor from the remainder of the design matrix. As can be seen in the above figure in the bottom panel, the variance inflation factor is approximately equal throughout the sequence timepoints being modeled that are used in RSA. The collinearity values for the FIR regressors of interest are well within the normal range that would enable us to obtain stable estimates of activity. These results indicate that our model is equally able to capture variance associated with early and late timepoints in the sequence (Mumford et al., 2015).

We have added the following sentences to the methods section and hope that our detailed vetting of our method satisfies this reviewer.

Methods Pg. 36:

“Collinearity in our model was measured using the variance inflation factor (VIF) and was verified to be within acceptable levels according to standard in the literature (Mumford et al., 2015).”

Minor:

- 6) In the methods, the authors describe the timeseries analysis as yielding “72 voxel timeseries” but I believe this is a typo

Response #6

We thank this reviewer for comment. Each sequence had 4 repetitions throughout the experiment, where one of them was a “catch” trial where the trial ended after position 3. These “catch” were excluded from both the cue period and navigation analyses, resulting in 3 repetitions of each sequence. See Clarke et al., 2021 for an empirical study examining these trials. There were 12 sequences in each context and 2 contexts which results in 72 unique trials for us to model using the FIR method discussed in the methods section. We added the following sentences to the methods section to make our analysis approach more clear:

Pg. 7 Results Section:

“In addition, one trial from each sequence was randomly chosen to end early at the rabbit (Catch Trials). This resulted in 72 sequences that could be analyzed.”

- 7) I thought it could be worth a Discussion sentence or two juxtaposing the current study results more directly with the Chanales... Kuhl 2017 paper, which the authors did cite. Despite some differing outcomes, there are a lot of structural similarities in the environment, and a small comment on this may prompt formal comparative research ideas.

Response #7

We agree that the work presented in Chanales et al., 2017 is highly relevant to the current study. We have now added the following paragraph to the discussion section juxtaposing this work and other related work that has structural similarities to our own.

Discussion Pg 21:

“Our findings are also relevant to past work that has examined how the brain represents routes with multiple paths or that are hierarchical in nature (Brown et al., 2014; Balaguer et al., 2016; Chanales et al., 2017). These studies show that activity in the hippocampus is higher when planning and navigating an overlapping route and that, during navigation, univariate bold signal

is modulated by distance to a goal. In one study, Chanales et al. (2017) show that representations of overlapping spatial routes become dissimilar over learning. This is potentially at odds with the current findings, where we find that routes that overlap in their goal show higher pattern similarity compared to routes that do not share a goal. However, participants in Chanales et al. (2017) passively viewed pictures along routes, whereas participants in our task actively navigated the state space. As mentioned earlier, rodent studies suggest that hippocampal spatial coding can shift dramatically between goal-directed behavior and random foraging in the same context. Moreover, in Chanales et al. (2017) it would make sense for participants to differentiate overlapping routes because they did not include sequences that converged on the same goal. Thus, it would be optimal to learn a unique representation for each spatial route in order to predict the outcome. In contrast, in our experiment, all trials that converged on the same goal required the same key decision at position 3, regardless of the starting point. In this situation, it is optimal to learn a representation that captures the information that is common to any sequence that converges on the same goal. For example, as depicted in Figure 1, any trial with a tiger as the goal animal will require participants to choose the “down” button at position 3. In the next section, we explain why results from the navigation period are also consistent with this interpretation.”

REVIEWER COMMENTS

Reviewer #1 (Remarks to the Author):

The revised manuscript is much improved and I commend the authors on their efforts. I particularly appreciate the thorough revision of the introduction, as well as the points of clarification and the additional analyses. The authors have addressed all of my concerns.

Reviewer #2 (Remarks to the Author):

Thank you for the detailed replies. All my questions/ requests have been addressed.

Reviewer #3 (Remarks to the Author):

I would like to thank the authors for their very thorough response to my comments and critiques. Overall, I feel the comments were fully and conscientiously addressed. Elements of the response such as the visualization of the collinearity structure for the FIR, and added juxtaposition with other work using similar designs, were quite useful.

I had a couple of minor comments for this revised manuscript:

1) In response to R3 comment #5, the authors did a nice job unpacking and explaining both the theoretical and methodological points related to their results and interpretations. Early in this response they noted the critical issue of the variable timing of cognitive processes themselves and how this may intersect with state overlap to obfuscate significant outcomes in the timecourse analysis. They offered to put some discussion of this in the main text or supplement if useful – I do think they raise a very important consideration for interpreting the results and comparing with other tasks, particularly since the timing of certain cognitive processes may differ between subjects, conditions, and different stages of experience with the task. I would recommend adding this to the discussion or supplement.

2) One small thing that caught my eye about this discussion point in the response letter was the statement that “previous fMRI studies have investigated representations of state spaces during incidental exposure to specific task states (e.g., during a task involving decisions about particular stimuli) or exposure to passively learned sequences with precise timing. To our knowledge, no prior fMRI studies have utilized active, self-initiated navigation through a state-space.” – I’m not sure I would characterize many of the prior efforts in this area as passive sequence learning tasks. Most that I am aware of require explicit, feedback-based acquisition of the routes based on route response errors and relationships between locations of interest. I note this more-so in case the authors choose to make a similar statement in revision, but not in disagreement of the overall point about pinning down neural representations associated with freely evolving cognitive states (which I think is a very important one)

REVIEWER COMMENTS (in black) and Responses (in blue)

We appreciate the reviewers' careful consideration of our manuscript and their thoughtful comments on how to improve our work. We have addressed all remaining comments and have incorporated these changes into our manuscript, as indicated below.

Reviewer #1 (Remarks to the Author):

- 1) The revised manuscript is much improved and I commend the authors on their efforts. I particularly appreciate the thorough revision of the introduction, as well as the points of clarification and the additional analyses. The authors have addressed all of my concerns.

Response #1

We thank this reviewer for their constructive comments and the opportunity to improve the manuscript.

Reviewer #2 (Remarks to the Author):

- 1) Thank you for the detailed replies. All my questions/ requests have been addressed.

Response #1

We would like to thank Reviewer #2 for their comments and appreciate their interest in our work.

Reviewer #3 (Remarks to the Author):

I would like to thank the authors for their very thorough response to my comments and critiques. Overall, I feel the comments were fully and conscientiously addressed. Elements of the response such as the visualization of the collinearity structure for the FIR, and added juxtaposition with other work using similar designs, were quite useful.

I had a couple of minor comments for this revised manuscript:

- 1) In response to R3 comment #5, the authors did a nice job unpacking and explaining both the theoretical and methodological points related to their results and interpretations. Early in this response they noted the critical issue of the variable timing of cognitive processes themselves and how this may intersect with state overlap to obfuscate significant outcomes in the timecourse analysis. They offered to put some discussion of this in the main text or supplement if useful – I do think they raise a very important consideration for interpreting the results and comparing with other tasks, particularly since the timing of certain cognitive processes may differ between subjects, conditions, and different stages of experience with the task. I would recommend adding this to the discussion or supplement.

Response #1

We agree with Reviewer 3's point that the timing of cognitive processes is critical for the interpretation of our time course similarity analyses. Like this reviewer mentioned, there are likely differences across trials within a participant and across participants for when a plan or action is initiated. One important distinction we would like to point out is that participants were trained to 85% criterion outside of the scanner. Because of this, we think that experience with the task has a negligible impact on the timing of the cognitive processes observed in our task. We have added the following text to the discussion section to highlight these points and help readers better interpret our results.

Added to discussion pages 23-24:

As noted above, the animals in the first three positions overlapped across diverging sequences, whereas the animals in the last three positions overlapped across converging sequences. Thus, if the hippocampus only represented the current state during navigation, we would have expected pattern similarity on the diagonal in Figure 4 to be higher for diverging trials for early time points, and then higher for converging trials in the later time points (see also Figure S4). If participants solely retrieved past states during navigation, we would expect off-diagonal pattern similarity to be higher for diverging sequences than converging sequences (because the first three positions were common for the diverging sequences). Our data were inconsistent with both of these accounts. Instead, we found that off-diagonal pattern similarity was higher for converging than for diverging trial pairs, suggesting that hippocampal activity patterns carried information about future timepoints during navigation.

The significant cluster of increased pattern similarity for converging, relative to diverging, sequences was consistent with the interpretation that, at the outset of the navigation phase, participants prospectively activated a representation of position 3. This result is notable for two reasons. First, participants were engaged in active, self-initiated navigation, and as such, we would expect considerable variability in the timing of prospective coding across trials and across subjects. The fact that prospective coding of position 3 (as indicated by off-diagonal pattern similarity) was nonetheless reliable across participants attests to the significance of this position to successful task performance. Second, the finding is notable because the stimulus at position 3 is exactly the same for all trials in all contexts. Thus, the disproportionate representation of position 3 across convergent sequences could not solely reflect the identity of the stimulus itself.

As noted above, the correct decision to be made at position 3 depends on one's current goal and context. All converging sequences share the same decision at position 3 because they share the same goal, whereas diverging sequences are associated with different decisions at position 3 because they involve different goal states. These results are consistent with the idea that participants prospectively activated the most goal-relevant information in the upcoming sequence, namely the context- and goal-appropriate decision at position 3.

Added to discussion page 27:

"We believe that hippocampal representations of physical space (Ekstrom and Ranganath, 2017) and abstract state spaces (Boorman, Sweigert, and Park, 2021) are flexible, reflecting the computational demands of the planning problem, and the subject's understanding of, and

experience with, the problem. In the present study, the task might have encouraged a model-based planning strategy in which future goals and key states are strategically retrieved and represented in hippocampus. In cases where learning is passive and incidental to the task, or when transitions between states change unpredictably, hippocampal state spaces might instead resemble successor-based maps. Finally, in more complex tasks, participants might adopt different strategies with varying degrees of emphasis on goal-relevant information (See Eldar et al., 2020).”

2) One small thing that caught my eye about this discussion point in the response letter was the statement that “previous fMRI studies have investigated representations of state spaces during incidental exposure to specific task states (e.g., during a task involving decisions about particular stimuli) or exposure to passively learned sequences with precise timing. To our knowledge, no prior fMRI studies have utilized active, self-initiated navigation through a state-space.” – I’m not sure I would characterize many of the prior efforts in this area as passive sequence learning tasks. Most that I am aware of require explicit, feedback-based acquisition of the routes based on route response errors and relationships between locations of interest. I note this more-so in case the authors choose to make a similar statement in revision, but not in disagreement of the overall point about pinning down neural representations associated with freely evolving cognitive states (which I think is a very important one)

Response #2

We thank the reviewer for this point and will clarify our reasoning here as we do not intend to add anything more relating to this point beyond what is discussed in Reviewer 3 Response #1. Characterizing past work as “passive sequence learning” is not an accurate description of past work and we note that there are several other works where state spaces have been employed (e.g. Schapiro et al., 2016, Kurth-Nelson et al., 2016, Constantinescu et al., 2016). However, these studies either employ passive learning, do not use fMRI, or do not report this effect. The purpose of this comment was to distinguish ourselves from past work and to illustrate the importance of studying cognitive processes as they freely evolve during behavior.